# Sub-network Discovery and Soft-masking for Continual Learning of Mixed Tasks

**Zixuan Ke**[1][*], **Bing Liu**[1], **Wenhan Xiong**[2], **Asli Celikyilmaz**[2] and **Haoran Li**[2]
[1]Department of Computer Science, University of Illinois at Chicago
[2]Meta
[1]{zke4,liub}@uic.edu
[2]{xwhan,aslic,aimeeli}@meta.com

## Abstract

Continual learning (CL) has two main objectives: preventing *catastrophic forgetting* (CF) and encouraging *knowledge transfer* (KT). The existing literature mainly focused on overcoming CF. Some work has also been done on KT when the tasks are similar. To our knowledge, only one method has been proposed to learn a sequence of mixed tasks. However, these techniques still suffer from CF and/or limited KT. This paper proposes a new CL method to achieve both. It overcomes CF by isolating the knowledge of each task via discovering a subnetwork for it. A soft-masking mechanism is also proposed to preserve the previous knowledge and to enable the new task to leverage the past knowledge to achieve KT. Experiments using classification, generation, information extraction, and their mixture (i.e., heterogeneous tasks) show that the proposed method consistently outperforms strong baselines.[1]

## 1 Introduction

One of the overarching goals of AI is to develop agents that can continually learn diverse tasks. Toward this goal, continual learning (CL) has been proposed, which incrementally learns a sequence of tasks, 1, ...., $T$ (Chen and Liu, 2018). Once a task $t$ is learned, its training data $D_t$ (at least a majority of it) is no longer accessible. This paper studies CL of NLP tasks (Ke and Liu, 2022), where a *task* is an *end-task*, e.g., text classification, summarization or information extraction, in the popular setting of *task-incremental learning* (TIL), in which the task-id is given in both training and testing.[2]

Ideally, CL should (1) overcome *catastrophic forgetting* (**CF**), i.e., it should not degrade the performance of previously learned tasks; (2) encour-

age *knowledge transfer* (**KT**) across tasks, i.e., the learning of a task should be able to leverage the knowledge learned from previous tasks[3]; and (3) be able to learn a mixed sequence of similar and dissimilar tasks and achieve both (1) and (2).

Most existing approaches for TIL address one of the three to the detriment of the others. For example, HAT (Serrà et al., 2018) and SupSup (Wortsman et al., 2020) can overcome CF by isolating the parameters of each task, which makes KT difficult. CUBER (Lin et al., 2022a) and DAS (Ke et al., 2023) can achieve KT by allowing some updates to existing knowledge, which causes CF. ProgressiveNet (Rusu et al., 2016) assures no CF by fixing all the learned parameters and expanding the network, but it limits KT. To our knowledge, CAT (Ke et al., 2020) is the only system that tries to have all three capabilities. However, it still suffers from CF due to its weak task similarity detection (for KT) and parameter sharing across dissimilar tasks.

This paper takes a major step to achieve all three objectives without suffering from the above drawbacks. The proposed system is called **TSS** (**T**IL based on **S**ub-network discovery and **S**oft-masking). TSS performs two functions, **(A)** sub-network discovery with soft-masking and **(B)** importance computation, to learn a mixed sequence, where some tasks may be *similar* (which enable KT) and some may be *dissimilar*.

Given a fixed and randomly initialized backbone network $N$, function **(A)** finds a separate sub-network as the model *for each task*. A sub-network is indicated by a set of *binary gates*, one for each parameter of $N$, specifying whether the corresponding parameter in $N$ should be in the sub-network for the task. In training, $N$ is fixed and the binary gates are obtained by learning a set of positive and real-valued *popup scores* (one for

---

[*]The work was done mainly during Meta AI internship.

[1]The code of TSS can be found at `https://github.com/ZixuanKe/PyContinual`.
[2]Another popular CL setting is *class-incremental learning* (CIL), which provides no task-id in testing and solves a different type of problems (Ke and Liu, 2022).

---

[3]This is called *forward transfer* in the literature. There is also the *backward transfer*, where the performance of previous tasks may be improved after learning a similar new task.

each parameter of $N$) and then applying a threshold on the popup scores.[4] In other words, *we convert the network training into the popup scores training.*[5] Only the binary gates (1's) obtained from the trained popup scores are stored for each task as its model. Thus, there is no interference across tasks to cause CF as the backbone network is fixed.

This sub-network discovery is effective for overcoming CF, but it prevents KT if the popup scores of each task are independently trained with no sharing across tasks. A naive way to share knowledge is to initialize the popup scores of the new task $t$ with the trained popup scores of the previous task $t - 1$. However, this is problematic for a mixed sequence of similar and dissimilar tasks because the training of task $t$ can only leverage the knowledge from the $(t-1)$th task. This can cause negative transfer if task $t-1$ is dissimilar to task $t$. To address this, we propose a **soft-masking** mechanism to make the initialization of popup scores for the new task contain the learned knowledge from *all* previous tasks so that the new task training can leverage the knowledge of all previous similar tasks. This is a **key novelty** of our approach. It is achieved by reducing (called "soft-masking") the gradient corresponding to the subset of popup scores that are **important** to previous tasks $(1...t-1)$ in training the current task $t$ so that the previously learned knowledge is well protected. In this way, the trained popup scores for task $t$, which will be used as the initialization for the next task, contain not only the knowledge for task $t$ but also the knowledge from all previous tasks. This is **critical** for learning a mixed sequence of similar and dissimilar tasks because the system does not know *a priori* which tasks' knowledge can be transferred to the current task. Note that soft-masking is only applied in backward propagation to preserve old knowledge. In the forward pass, the training can still use all popup scores corresponding to all parameters of the backbone network. Additionally, soft-masking reduces the gradient instead of fully blocking the gradient, which gives the model the flexibility to *adjust* any popup scores when training the current task $t$, which further encourages KT.

The question is how to decide the important popup scores for each task. This is done by **(B)**,

which computes the importance of the popup scores *after* training a task. The importance is a number between 0 and 1. After training popup scores for a task, we input the training data *again* to compute the importance of each popup score for the task based on its gradient. We save the *accumulated* importance over all previous tasks to keep track of the important popup scores so far. The accumulated importance (within 0 and 1; thus "soft") is used to guide soft-masking in **(A)** in training a new task.

This paper makes three key contributions.

1. It studies an important but under-studied problem of *continually learning* a mixed sequence, where some tasks may be similar and some may be dissimilar. Existing CL methods suffer from either CF or limited KT in this scenario.

2. It proposes a novel method TSS consisting of sub-network discovery with soft-masking and importance computation to prevent CF and to encourage KT for a mixed sequence of tasks.

3. It evaluates TSS using datasets consisting of classification, generation and information extraction tasks and their mixture. The results demonstrate the effectiveness of TSS.

## 2    Related Work

**Forgetting prevention in continual learning (CL).** There are four main families of approaches:

(1) *Regularization-based approaches* (Kirkpatrick et al., 2016; Lee et al., 2017; Seff et al., 2017; Zenke et al., 2017; Rusu et al., 2016) add a regularization in the loss to penalize changes to parameters that are important to previous tasks. *Gradient projection* (Zeng et al., 2019) ensures the gradient updates occur in the orthogonal direction to the input of old tasks and in trust region (Lin et al., 2022a,b) TSS uses no regularization.

(2) *Replay-based approaches* (Rebuffi et al., 2017; Lopez-Paz and Ranzato, 2017; Chaudhry et al., 2019; Wang et al., 2020) retain some training data of old tasks and use them in learning a new task. The methods in (Shin et al., 2017; Kamra et al., 2017; Rostami et al., 2019; He and Jaeger, 2018) learn data generators and generate old task data for learning a new task. These are clearly different from TSS as it uses no any replay data.

(3) *Parameter isolation* (Serrà et al., 2018; Ke et al., 2020, 2021a; Mallya and Lazebnik, 2018; Fernando et al., 2017; Wortsman et al., 2020) learns and masks a dedicated sub-network for each task, but has limited KT. TSS leverages this approach to

---

[4] The popup score means that the score is used to select a parameter or popup the edge from the backbone network (Ramanujan et al., 2020).

[5] Therefore, training task $t$ is the same as training the popup scores for task $t$. We thus use the two terms interchangeably.

isolate knowledge for different tasks, but enables KT by using a soft-mask mechanism to preserve the learned knowledge and uses the old knowledge as the initialization for learning the new task.

(4) *Parameter isolation plus out-of-distribution (OOD) detection* is a new and theoretical grounded approach (Kim et al., 2022, 2023). It is mainly used for class-incremental learning (CIL). The main idea is to use a parameter isolation approach for task incremental learning (TIL) to overcome CF and the model for each task is an OOD detection model. During inference, the system performs both task-id prediction and within-task prediction for CIL classification. However, our work is about TIL.

**Knowledge transfer in CL.** There are two main approaches for KT: (1) *replay-based* (Huang et al., 2021; Wang et al., 2021, 2020; de Masson d'Autume et al., 2019; Zhu et al., 2022; Yin et al., 2022), which use replay data to help the transfer. TSS is replay-free. (2) *similarity-based*, which uses task similarities using features (Ke et al., 2021a,b, 2020; Wang et al., 2022; Zhang et al., 2022) or gradients (Lin et al., 2022a). However, these methods are not always reliable, which results in CF. CAT (Ke et al., 2020) considers a mixed sequence of tasks by first detecting previous tasks that are similar to the current task and then opening the masks of these tasks so that the new task learning can modify their parameters to achieve KT. However, this causes CF for dissimilar tasks that share parameters with those similar tasks. DAS (Ke et al., 2023) encourages KT by allowing previous knowledge updates based on importance, but this also causes CF because previous parameters is changeable and there is no easy way to separate the parameters that are used for different tasks. In contrast, TSS does not require any explicit similarity detection but finds sub-network for each task to guarantee no CF. Its soft-masks allow the initialization contains all previously learned knowledge and thus encourage KT to the new task. Konishi et al. (2023) recently proposed a parameter-level soft-masking method using AlexNet as the backbone. The method has difficulties working with more complex architectures. Our soft-masks are set on the popup scores rather than network parameters and our approach is based on sub-network discovery and knowledge sharing.

**Network pruning as importance computation.** It is known that many parameters in a neural network are redundant and can be pruned (Li et al., 2021; Lai et al., 2021; Chen et al., 2020; Lin et al., 2020; Gao et al., 2021; Voita et al., 2019). For Transformer, one can prune the attention head (Michel et al., 2019; Voita et al., 2019; McCarley et al., 2019) and sub-layers (Fan et al., 2020; Sajjad et al., 2020). However, these methods are not directly applicable to us as we need to compute the importance of each popup score instead of each parameter in the network. We use the importance as soft-masks to leverage all existing sub-networks for KT rather than to compress the LM.

## 3 Proposed TSS Technique

TSS is designed for both CF prevention and forward KT. Figure 1 gives an overview of TSS. The backbone network consists of the transformer and adapters, which are indicated by the transformer parameters $w_l^{\text{transformer}}$ of layer $l$ and by the parameters of adapter $w_l^{\text{adapter}}$ of layer $l$ respectively (see the grey boxes). Note again that these are fixed in the entire learning process. On the left, we show the forward pass and backward propagation of **(A) sub-network discovery with soft-masking** (Sec. 3.1). In the forward pass, a set of learnable popup scores $s_l^{(t)}$ are initialized (Sec. 3.1.2) and fed into a step function. The output of the step function is a set of binary gates $g_l^{(t)}$ that element-wise multiply ($\otimes$) with the parameters of the adapter of the backbone network ($w_l^{\text{adapter}}$). As a result, the binary gates indicate a sub-network for task $t$ within the $w_l^{\text{adapter}}$ (Sec. 3.1.1). In the backward propagation (Sec. 3.1.2), we first accumulate the importance of popup scores for all previous tasks by normalization and element-wise maximum ("EMax"). The accumulated importance $I_l^{(\leq t-1)}$ is converted into soft-mask by the operation $1 - I_l^{(\leq t-1)}$. This soft-mask is then element-wise multiplied with the original gradient of the popup scores $s_l^{(t)}$ in layer $l$, $\nabla_l$ (computed by straight-through estimator in Eq. 3). The soft-masked gradient, $\hat{\nabla}_l$, is the final gradient that is used to optimize the popup scores $s_l^{(t)}$. Since the important popup scores have lower gradients in $\hat{\nabla}_l$, the popup scores that are important to previous tasks are preserved and are used to initialize the new task to encourage KT (not shown in Figure 1).

After **(A)**, we show **(B) importance computation** (Sec. 3.2). The forward pass is the same as **(A)**. However, in backward propagation, the gradient of popup scores, $\nabla_l$, is not used to update anything (red cross in the arrow) but computes

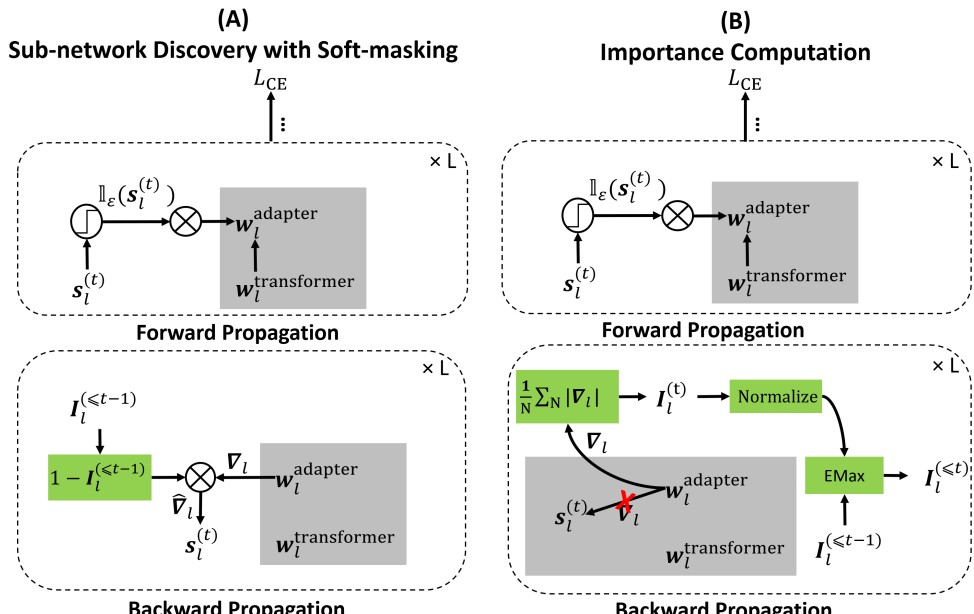

**(A)**
**Sub-network Discovery with Soft-masking**

**(B)**
**Importance Computation**

Figure 1: Illustration of TSS in training task $t$. The detailed description is in Sec. 3. The grey boxes indicate that the parameters of the adapters and backbone language model (LM) are all frozen. The only trainable parameters are the popup scores $s_l^{(t)}$. In **(A) sub-network discovery with soft-masking** (Sec. 3.1.1), we remove the step function in the backward propagation to reflect the straight-through estimator in Eq. 3, where the gradient "skips" the step function as if it is an identity function. The original gradient for popup scores, $\nabla_l$, is not directly used to update the popup scores but soft-masked based on the accumulated importance (Sec. 3.1.2). In **(B) Importance Computation** (Sec. 3.2), the red cross in backward propagation indicates that the gradient is not used to update the popup scores $s_l^{(t)}$ but only to compute the importance.

the importance using its absolute average value, $\frac{1}{N} \sum_{n=1}^{N} |\nabla_l|$. The resulting importance of task $t$ in layer $l$, $I_l^{(t)}$, is saved for **(A)** in learning the next task. The following sub-sections present each function and step in detail.

### 3.1 Sub-network Discovery and Soft-masking

Sub-network discovery finds a sub-network (represented as binary gates) for the current task, which is saved to overcome CF. It relies on the training of popup scores and its training also involves soft-masking to encourage knowledge transfer. In this section, we will present how we discover the sub-network via learning popup scores and how we encourage knowledge transfer (KT) via soft-masking.

#### 3.1.1 From Popup Scores to Sub-network

Since the pre-trained language model (LM) contains useful general knowledge, we fix and use the full LM. The sub-network discovery is only done on adapters (Houlsby et al., 2019), which are inserted into each layer of the LM.[6] Note again that throughout the training process, both the adapters (randomly initialized) and backbone LM are *fixed*.

We now define a set of learnable **popup scores**

---

[6]See Appendix A for additional details about adapters.

$s_l^{(t)}$ for layer $l$ for task $t$. It has the same size as the parameters of the adapters $w_l^{\text{adapter}}$ (which is fixed throughout learning). In the forward pass, $s_l^{(t)}$ is constantly *thresholded*,

$$g_l^{(t)} = \mathbb{1}_\epsilon(s_l^{(t)}), \qquad (1)$$

where $\mathbb{1}$ is a step function that outputs 1 for the scores that are larger than the threshold $\epsilon$ (a hyperparameter). Since the gate $g_l^{(t)}$ is always binary, it can naturally indicate a **sub-network** within the parameters of the fixed adapter, $w_l^{\text{adapter}}$, by element-wise multiplication,

$$\hat{w}_l^{\text{adapter}} = w_l^{\text{adapter}} \otimes g_l^{(t)}, \qquad (2)$$

where $\otimes$ is element-wise multiplication. $\hat{w}_l^{adapter}$ indicates the *selected* parameters in the adapter that form the discovered sub-network.

**Training the popup scores.** $s_l^{(t)}$ cannot be directly optimized because the derivative of its threshold-based step functions is zero. Then nothing can be learned. To solve the problem we use straight-through estimators (Bengio et al., 2013). Specifically, we ignore the derivative of the threshold function and pass on the incoming gradient as if the threshold function was an identity function.

Consequently, for each parameter $s_{i,j}^{(t)}$ in $\boldsymbol{s}_l^{(t)}$, it is updated as follows,

$$\boldsymbol{s}_l^{(t)} = \boldsymbol{s}_l^{(t)} - \alpha \nabla_l;$$
$$\nabla_l = \frac{\partial \mathcal{L}}{\partial s_{i,j}^{(t)}} = \frac{\partial \mathcal{L}}{\partial \mathcal{I}_j} \frac{\partial \mathcal{I}_j}{\partial s_{i,j}^{(t)}} = \frac{\partial \mathcal{L}}{\partial \mathcal{I}_j} w_{i,j} \mathcal{O}_i, \qquad (3)$$

where $\mathcal{I}_j$ and $\mathcal{O}_i$ are the input and output of neurons $i$ and $j$. $\nabla_l$ is the gradients on the popup scores $\boldsymbol{s}_l^{(t)}$ and $\alpha$ is the learning rate. After training, the binary gate, $\boldsymbol{g}_l^{(t)}$, is saved (taking only 1-bit per element). Since $\boldsymbol{g}_l^{(t)}$ is saved, and all the backbone parameters are fixed (including the adapter's and backbone LM's), TSS does not suffer from forgetting of the previously learned knowledge.

### 3.1.2 Preserving Previous Knowledge for New Task Initialization

While the sub-network discovery in Sec. 3.1.1 can help prevent forgetting (CF), it does not do knowledge transfer (KT). As discussed earlier, the naive way for KT is to initialize the popup scores $\boldsymbol{s}_l^{(t)}$ for the new task $t$ with the learned scores $\boldsymbol{s}_l^{(t-1)}$ for task $t-1$. However, $\boldsymbol{s}_l^{(t-1)}$ contains only the knowledge from task $t-1$, but we have no idea whether any knowledge from task $t-1$ can be transferred to task $t$ because tasks $t$ and $t-1$ may be entirely different. To address this, we preserve the learned knowledge from all previous tasks from 1 to $t-1$ in $\boldsymbol{s}_l^{(t-1)}$. In this way, task $t$ can leverage all previously learned knowledge for KT.

To make the preservation possible, inspired by (Ke et al., 2023), we propose a soft-masking mechanism based on the importance of each popup score to all previous tasks. We compute the importance in Sec. 3.2. For now, we assume that we already have the set of importance value $\{\boldsymbol{I}_l^{(k)}\}_{k=1}^{t-1}$ for $\boldsymbol{s}_l^{(k)}$ for each previously learned task $k$. The preservation is achieved by soft-masking the learning based on the accumulated importance as follows:[7]

**Accumulating importance.** We accumulate the importance after task $t-1$ is learned via element-wise max (EMax),

$$\boldsymbol{I}_l^{(\leq t-1)} = \text{EMax}(\{\boldsymbol{I}_l^{(t-1)}, \boldsymbol{I}_l^{(\leq t-2)}\}), \qquad (4)$$

---

[7]Before accumulation, we normalized the the importance values in each layer $l$ for a task $k$ by making the importance values for all popup scores in the layer to have the mean of 0 and standard deviation of 1. To further facilitate soft-masking, the normalized importance values are rounded by a Tanh activation and the absolute operation so that the values are in the interval of [0,1]. To simplify the notation, we still use $\boldsymbol{I}_l^{(k)}$ to represent the resulting importance.

where $\boldsymbol{I}_l^{(\leq t-2)}$ refers to the previously accumulated importance at task $t-2$. We do not need to save all $\{\boldsymbol{I}_l^{(k)}\}_{k=1}^{t-1}$ for Eq. 4, but only the incrementally accumulated importance after training each task.

**Soft-masking the popup scores.** Given the accumulated importance $\boldsymbol{I}_l^{(\leq t-1)}$ of layer $l$ and the cross-entropy loss $\mathcal{L}_{\text{CE}}$, we reduce (or soft-mask) $\boldsymbol{s}_l^{(t)}$'s gradient ($\nabla_l$ in Eq. 3) flow as follows,

$$\hat{\nabla}_l = (1 - \boldsymbol{I}_l^{(\leq t-1)}) \otimes \nabla_l, \qquad (5)$$

The importance $\boldsymbol{I}_l^{(\leq t-1)}$ has the same size as $\nabla_l$ and the associated popup scores. This is *soft-masking* as each element in $\boldsymbol{I}_l^{(\leq t-1)}$ is a real number in $[0, 1]$ (not binary $\{0, 1\}$), which gives the model the flexibility to adjust any popup scores. We note that the above soft-masks are only applied in the backward pass, but not in the forward pass, which encourages knowledge transfer because each task training can leverage all popup scores that are corresponding to all parameters of the backbone network and the popup scores contain the knowledge from all previously learned tasks.

**Popup scores initialization.** Thanks to the soft-masking mechanism, the trained $\boldsymbol{s}_l^{(t-1)}$ contains knowledge from all previous tasks. We use it to initialize the popup scores for the new task $\boldsymbol{s}_l^{(t)}$ to encourage knowledge transfer. Note that we apply Kaiming Initialization for $\boldsymbol{s}_l^{(0)}$.

### 3.2 Computing the Importance of Popup Scores to the Current Task

To apply soft-masking (Sec. 3.1.2), we need to determine the importance of popup scores for the previous tasks. This is done after training a task. We adapt the gradient-based importance detection method in (Michel et al., 2019) for our purpose. Given a dataset $D = \{(\boldsymbol{x}_i, y_i)\}_{i=1}^n$ of $n$ samples ($y_i$ is the class label of $\boldsymbol{x}_i$) and the trained popup scores $\boldsymbol{s}_l^{(t)}$, we input the data *again* and the importance of pupup scores in the layer $l$ is estimated with a gradient-based proxy score

$$\boldsymbol{I}_l^{(t)} = \frac{1}{N} \sum_{n=1}^{N} \frac{\partial \mathcal{L}_{\text{CE}}(\boldsymbol{x}_n, y_n))}{\partial_{\boldsymbol{s}_l^{(t)}}}, \qquad (6)$$

Note the average gradient computed in Eq. 6 over all the data is only used to compute the importance and will not be used to optimize the popup scores. The resulting $\boldsymbol{I}_l^{(t)}$ is of the same size as $\boldsymbol{s}_l^{(t)}$, each entry corresponding to the importance of a popup

score. It is used in the next task by accumulating with the previously accumulated importance (Eq. 4) and soft-masking the learning (Eq. 5). Note that $I_l^{(0)}$ is all 0 as we do not know which popup scores are important before the training of the first task.

# 4 Experiments

We now evaluate the proposed system TSS. We first learn all tasks sequentially. After that, their task models are tested using their respective test sets. TSS does not use any replay data.

## 4.1 Datasets and Baselines

**Datasets:** We use five datasets covering a wide range of NLP problems, including classification, generation, and information extraction. For detailed datasets statistics, please see Appendix B. **(1) ASC** (*Aspect Sentiment Classification*) is from (Ke et al., 2021a) and has 19 tasks. Each task classifies the opinion (*positive*, *negative*, or *neutral*) in a review sentence at the aspect-level of a product. **(2) CCD** (*Continual Classification Dataset*) is a popular continual text classification dataset (de Masson d'Autume et al., 2019).[8] **(3) SUM** (*ConvoSum*) is a conversational abstractive summarization dataset with 6 tasks/domains (Fabbri et al., 2021). Given conversations from a domain, the system generates its summary. **(4) DRG** (*Dialogue Response Generation*) is a popular task-oriented dialogue response dataset (Multi-WoZ2.0) (Ramadan et al., 2018) with 5 tasks/domains. Given the intent and dialogue state (slot-value pairs containing messages to express), the system generates a response. **(5) NER** (*Named Entity Recognition*)[9] classifies mentions into pre-defined entity types in each task.

Since TSS aims at (1) preventing forgetting, (2) encouraging knowledge transfer and (3) learning a mixed sequence of similar and dissimilar tasks, we consider two types of sequences.

**Homogeneous tasks sequences.** In each such task sequence, all tasks are from the same dataset. Among the aforementioned 5 datasets, two of them

(ASC and NER) are datasets consisting of **similar tasks** as the tasks share similar task labels (with some exceptions, see the statistic in Appendix B) but from different domains. Our goal is to achieve both CF prevention and KT. Three of them (SUM, CCD, and DRG) are **dissimilar tasks** as the distribution shift across tasks is relatively large. They have little shared knowledge to transfer and the main goal is to ensure there is little or no CF.

**Heterogeneous tasks sequence.** This sequence is constructed by mixing all the above similar and dissimilar tasks of different types from all the 5 datasets in random order. It has a total of **40** tasks. This is a challenging and more realistic setting where the system needs to prevent CF and encourage KT dynamically.

**Baselines.** We use **14** baselines with both *non-continual* and *continual learning* (CL) methods.

*Non-CL baselines*: **MTL** and **MTL (Adapter)**[10] train tasks in a multi-task or data combined setting, where the former trains the whole LM and the latter trains only the adapter. These two are widely accepted as the ***upper bounds*** of CL. **ONE** builds a separate model for each task by fine-tuning the LM, which has no KT or CF. **ONE (Adapter)** (Madotto et al., 2020) trains an adapter for each task separately (called AdapterCL in its original paper). **ONE (Prompt)** (Zhu et al., 2022) trains a prompt for each task (called C-PT in its original paper).

*CL baselines*. The CL baselines include an *naive continual learning* (NCL) method where the system learns the tasks one by one with no mechanism to deal with CF or to encourage KT, and **9** state-of-the-art TIL methods: . **5** adapter-based methods: **CTR** (Ke et al., 2021a), **HAT** (Serrà et al., 2018), **SupSup** (Wortsman et al., 2020), **CAT** (Ke et al., 2020) and **CUBER** (Lin et al., 2022a). **1** prompt-based method: **L2P** (Wang et al., 2022). **3** baselines that modify the Transformer: **LAMOL** (Sun et al., 2020), **EWC** (Kirkpatrick et al., 2016) and **DAS** (Ke et al., 2023). Readers can refer to Appendix D for the details of these baselines.

**LM and hyperparameters.** Since we need a backbone LM that can do classification, generation,

---

[8]It contains 5 tasks: AGNews (news classification), Yelp (sentiment analysis), Amazon (sentiment analysis), DBpedia (Wikipedia article classification) and Yahoo (questions and answers categorization). Since each of these datasets is quite large, we randomly sampled 500 samples from each class for each task due to our resource limitations.

[9]This data consists of 5 tasks, including **conll03** (Sang and Meulder, 2003), **wikigold** (Balasuriya et al., 2009), **btc** (Derczynski et al., 2016), **re3d** (Laboratory, 2017), and **gum** (Zeldes, 2017). Due to resource limitations, we randomly sampled 200 samples for each task.

[10]For classification datasets (ASC, CCD and NER), we conduct a multi-task learning (MTL) experiment. For generation datasets (SUM and DRG), MTL is not possible as the language modeling head on top of BART is a linear layer with weights tied to the input embeddings. We follow the standard practice (e.g., (Qin and Joty, 2022; Madotto et al., 2020)) and pool all data together to train a single shared head (we still called this MTL for simplicity).

| | Similar | | Dissimilar | | | Average | Average | Similar | | Dissimilar | | | Average | Average |
|---|---|---|---|---|---|---|---|---|---|---|---|---|---|---|
| Dataset | ASC | NER | SUM | CCD | DRG | Main | FR | ASC | NER | SUM | CCD | DRG | Main | FR |
| Model | MF1 | F1 | R1 | MF1 | BLEU | | | MF1 | F1 | R1 | MF1 | BLEU | | |
| **Non-continual learning** | | | | | | | | | | | | | | |
| MTL | 92.28 | 63.33 | 39.39 | 90.57 | 25.29 | 62.17 | — | 92.28 | 63.33 | 39.39 | 90.57 | 25.29 | 62.17 | — |
| MTL (Adapter) | 92.17 | 60.61 | 38.84 | 91.09 | 24.50 | 61.44 | — | 92.28 | 63.33 | 39.39 | 90.57 | 25.29 | 62.17 | — |
| ONE | 85.55 | 59.33 | 39.07 | 91.07 | 24.14 | 59.83 | — | 85.55 | 59.33 | 39.07 | 91.07 | 24.14 | 59.83 | — |
| ONE (Adapter) | 83.95 | 56.69 | 38.90 | 90.78 | 23.42 | 58.75 | — | 83.95 | 56.69 | 38.90 | 90.78 | 23.42 | 58.75 | — |
| ONE (Prompt) | 76.46 | 45.90 | 30.67 | 86.23 | 12.67 | 50.39 | — | 76.46 | 45.90 | 30.67 | 86.23 | 12.67 | 50.39 | — |
| **Continual learning of heterogeneous tasks** | | | | | | | | **Continual learning of homogeneous tasks** | | | | | | |
| NCL | 88.05 | 34.56 | 22.86 | 81.86 | 8.68 | 47.20 | 13.94 | 89.25 | 49.19 | 32.68 | 85.08 | 22.31 | 55.70 | 6.42 |
| EWC | 88.73 | 28.75 | 18.65 | 83.24 | 7.99 | 45.47 | 13.43 | 88.36 | 51.76 | 32.64 | 87.27 | 18.30 | 55.66 | 5.53 |
| HAT | 87.45 | 50.16 | 35.21 | 89.85 | 20.98 | 56.73 | 0.97 | 89.33 | 52.31 | 37.11 | 90.21 | 21.47 | 58.08 | 0.49 |
| DAS | 90.62 | 43.38 | 25.45 | 82.66 | 15.21 | 51.47 | 10.49 | 90.94 | 42.12 | 31.31 | 87.97 | 22.25 | 54.92 | 9.26 |
| SupSup | 85.83 | 58.93 | 38.23 | 90.66 | 24.71 | 59.67 | 0.00 | 85.83 | 58.93 | 38.23 | 90.66 | 24.71 | 59.67 | 0.00 |
| LAMOL | — | — | — | — | — | — | — | 84.62 | — | 10.88 | 54.44 | 19.96 | — | — |
| CAT | 45.79 | 35.24 | 17.44 | 36.62 | 10.21 | 29.06 | 30.27 | 84.31 | 50.73 | 37.24 | 90.82 | 21.72 | 56.96 | 1.37 |
| CTR | 89.37 | 50.16 | 33.98 | 89.96 | 17.63 | 56.22 | 0.95 | 88.86 | 51.85 | 37.34 | 90.54 | 21.39 | 58.00 | 0.65 |
| CUBER | 90.58 | 46.67 | 30.94 | 81.54 | 15.68 | 53.08 | 7.27 | 91.25 | 47.19 | 30.44 | 89.19 | 21.66 | 55.95 | 5.82 |
| L2P | 78.10 | 39.03 | 28.34 | 72.35 | 4.55 | 44.47 | 6.44 | 74.81 | 44.22 | 26.65 | 85.35 | 8.52 | 47.91 | 2.58 |
| TSS | **90.61** | **62.13** | **38.29** | **90.70** | **24.56** | **61.26** | **0.00** | **91.28** | **63.96** | **38.39** | **90.89** | **24.75** | **61.85** | **0.00** |

Table 1: Performance for **heterogeneous** tasks (a sequence of 40 tasks from all 5 datasets) and **homogeneous** tasks (each sequence consisting of all tasks from one dataset), averaged over 5 random sequences (the **standard deviations** are given in Appendix H due to space limits). "—" means not applicable. We bold the best results within CL baselines. The smaller forgetting rate (FR) means the system can deal with forgetting better. Note, the results for **non-continual learning** are the same in both settings. **Execution time** and **memory need** are given in Appendix J.

and information extraction. We adopt BART$_{\text{LARGE}}$ (Lewis et al., 2020) as our LM. Fine-tuning of BART follows the standard practice.[11] Detailed **hyperparameters** are given in Appendix C.

### 4.2 Evaluation Results and Analysis

Since **the order of the tasks** in a sequence may impact the final result, we ran 5 randomly sampled task sequences (results of individual sequences are given in Appendix I). For different types of tasks, we use their standard evaluation metrics.[12] Table 1 gives the average result of each system over 5 random task sequences *after* continual learning of **heterogeneous** task sequences (left section) and **homogeneous** task sequences (right section). We report the **main** performance of each dataset, i.e., MF1 (Macro-F1) in ASC and CCD, F1 in NER, R1 in SUM and BLEU in DRG and their average in the **Average Main** column (the second from the right in each section). We also report the average forgetting rate (FR) on the same metrics in the **Average FR** column to evaluate the forgetting prevention.[13]

LAMOL is not included in heterogeneous tasks as it is not obvious how to adapt LAMOL for NER.

**Heterogeneous Tasks.** The left section in Table 1 shows that TSS outperforms all CL baselines for the mixed sequence of 40 heterogeneous tasks, with similar tasks (from ASC, NER) and dissimilar tasks (from SUM, CCD and DRG). Note, although we have only one task sequence, we report the results separately for each dataset. Other observations are:

(1). **TSS is more effective than CL baselines that only deal with CF** (EWC, HAT, Sup-Sup). In the two similar datasets (ASC and NER), TSS clearly wins because regularization-based EWC sacrifices accuracy for overcoming CF and parameter-isolation based SupSup prevents any possible KT. In the 3 dissimilar datasets which have little shared knowledge to transfer, TSS is similar to the baseline that can prevent CF (like SupSup). This confirms TSS can achieve both CF prevention and KT in the challenging heterogeneous setting.[14]

(2). **TSS is more effective than CL baselines that deal with both CF and KT** (CAT, CTR, DAS, CUBER, and L2P). Among these systems, CAT performs the worst due to its inaccurate task similarity detection and its difficulty to deal with CF. L2P performs better, but still has a large gap to TSS due to the poor prompt selection (we can see it is

---

[11]For ASC, we adopt the ASC formulation in (Xu et al., 2019), where the aspect term and sentence are concatenated via . Opinion is predicted as the average over all tokens.

[12]We use Macro-F1 and accuracy for the sequence-level classification tasks (ASC and CCD), where Macro-F1 (MF1) is the primary metric because highly imbalanced classes in ASC introduce biases in accuracy. We use Rouge score (R1, R2 and RL) for SUM, BLEU score for DRG and F1 for NER.

[13]For the detailed metrics and results for each system, please see Appendix E. For the detailed forgetting rate for each dataset, please see Appendix G.

[14]Other baselines perform poorly: HAT has little KT in classification tasks, which makes ASC poorer. It has forgetting in generation tasks as it cannot isolate parameters in the shared LM head.

| Dataset | Similar | | Dissimilar | | | Average | Average | Similar | | Dissimilar | | | Average | Average |
| Model | ASC | NER | SUM | CCD | DRG | Main | FR | ASC | NER | SUM | CCD | DRG | Main | FR |
| | MF1 | F1 | R1 | MF1 | BLEU | | | MF1 | F1 | R1 | MF1 | BLEU | | |
|---|---|---|---|---|---|---|---|---|---|---|---|---|---|---|
| **Non-continual learning** | | | | | | | | | | | | | | |
| ONE | 85.55 | 59.33 | 39.07 | 91.07 | 24.14 | 59.83 | — | 85.55 | 59.33 | 39.07 | 91.07 | 24.14 | 59.83 | — |
| **Continual learning of heterogeneous tasks** | | | | | | | | **Continual learning of homogeneous tasks** | | | | | | |
| TSS (w/o SD) | 86.54 | 37.03 | 28.04 | 78.29 | 14.85 | 48.95 | 9.02 | 91.06 | 40.47 | 30.42 | 88.34 | 22.77 | 54.61 | 9.61 |
| TSS (w/o SM) | 85.83 | 58.93 | 38.23 | 90.66 | 24.71 | 59.67 | 0.00 | 85.83 | 58.93 | 38.23 | 90.66 | 24.71 | 59.67 | 0.00 |
| TSS (w/o SM; Naive) | 88.89 | 34.79 | 36.11 | 89.99 | 24.19 | 54.80 | 0.00 | 90.38 | 62.42 | 38.08 | 90.63 | 24.59 | 61.22 | 0.00 |
| TSS | **90.61** | **62.13** | **38.29** | **90.70** | **24.56** | **61.26** | **0.00** | **91.28** | **63.96** | **38.39** | **90.89** | **24.75** | **61.85** | **0.00** |

Table 2: Ablation experiment results for **heterogeneous** and **homogeneous** tasks - averages over 5 random sequences (the standard deviations are reported in Appendix H due to space limits).

even poorer than ONE (prompt), indicating that its selection causes CF). DAS performs well on ASC, but poorly on other datasets, indicating it cannot effectively prevent CF. CUBER has strong performance on similar tasks (e.g., ASC) but performs poorly on dissimilar tasks. CTR is the best among the three, but it is still worse than TSS due to its inaccurate instance-level similarity detection.

**Knowledge transfer (KT) and CF prevention.** To validate TSS's effectiveness in dealing with CF with a sequence of dissimilar tasks, we can first compare TSS with ONE. We can see TSS achieves similar results to ONE in the three **dissimilar tasks** datasets, indicating effective CF prevention. Additionally, we can evaluate the continual learning process to see whether forgetting occurs during the training of each system. To this end, we compute **Forgetting Rate**[15] in Table 1 (the right-most column in the CL of heterogeneous tasks section). Clearly, TSS has a 0 forgetting. SupSup also has 0 forgetting because it also tries to find a sub-network for each task. While this is certainly good for CF prevention but makes KT impossible. We can see other baselines all suffer from forgetting (positive forgetting rate) on average.

Regarding **KT**, we can again use ONE as the control and see whether there is effective KT (learned tasks help the new task). Clearly, we can see TSS outperforms ONE in two datasets (ASC and NER) with **similar tasks**, indicating effective KT. Thanks to the sub-network discovery and soft-masking, we can also see TSS is very similar to MTL/Comb, which again shows the effectiveness of TSS.

**Ablation study.** We want to know whether (1) the sub-network discovery and (2) the soft-masking

mechanism are effective. For (1), we conduct the ablation experiment **TSS (w/o SD)**, where we remove the sub-network discovery and directly train the parameters of adapters with soft-masking. For (2), we conduct the experiment **TSS (w/o SM)** and **TSS (w/o SM; Naive)**. They both do not use the soft-masking mechanism. The former initializes the popup scores with Kaiming Initialization for all tasks while the latter initializes the scores with only those of the last task.

The right section (heterogeneous) in Table 2 shows the ablation results and the corresponding forgetting rates. The full TSS gives the best average result, indicating that every component helps. We further observe: (1) TSS's gain is partially from the sub-network discovery as TSS (w/o SD) is poorer on average, particularly for those datasets having little shared knowledge; (2) soft-masking helps as TSS (w/o SM) gives a worse performance; (3) soft-masking can help make the initialization for the new task better as TSS (w/o SM; Naive) is clearly worse than TSS. Note that both TSS (w/o SM) and TSS (w/o SM; Naive) have 0 forgetting rate, due to the effectiveness of sub-network discovery.

**Homogeneous Tasks.** For continual learning of homogeneous tasks, we conducted 5 experiments, one for each for the 5 datasets, i.e., each task sequence contains only the same type of tasks (see Sec. 4.1). The right section of Table 1 shows the average results for each system. The observations are similar to those for heterogeneous tasks, i.e., TSS outperforms all CL baselines. We further observe that TSS has larger improvements on similar tasks (ASC and NER), comparing to the improvements in the mixed sequence. This is expected.

The right section of Table 2 shows the ablation study. The observations are also similar to those for heterogeneous tasks, i.e., every component helps. We also observe that the naive transfer (TSS (w/o SM; Naive)) works similarly to TSS. This indicates that the proposed initialization is more important

---

[15]The forgetting rate (Liu et al., 2020) is defined as FR $= \frac{1}{t-1} \sum_{i=1}^{t-1} A_{i,i} - A_{t,i}$, where $A_{i,i}$ is the test performance of each task when it was first learned and $A_{t,i}$ is the performance of task $i$ after training the last task $t$. We average over all tasks except the last one as the last task obviously has no forgetting. The detailed forgetting rate for each dataset is given in Appendix G. (Mehta et al., 2021) defined a different forgetting rate. Appendix F will argue that ours is more effective.

in the heterogeneous tasks setting because two adjacent tasks can be very dissimilar (e.g., the current task is NER but its previous/last task may be summarization) and causes negative transfer.

In summary, we can say that TSS works well for both the homogeneous tasks and the challenging heterogeneous tasks scenarios.

## 5 Conclusion

This paper studied *task incremental learning* (TIL) on a range of NLP problems. We first presented the three desired capabilities: no forgetting, knowledge transfer and learning a mixed sequence of similar and dissimilar tasks. To our knowledge, only one system (CAT) aims to achieve all these objectives but it suffers from forgetting. We then propose a novel method, TSS, to achieve all three. Experimental results showed that TSS achieves KT and no forgetting even for the challenging mixed (or heterogeneous) task sequence setting.

## 6 Limitation

While effective, TSS has some limitations. First, although TSS shows strong performance in forward transfer (old knowledge helps the new task), how to enable backward transfer (new knowledge improves the trained tasks) is left to future work. Second, while empirically soft-masking the gradient does not harm task learning, there is no theoretical guarantee that the task with soft-masking can learn as well as without soft-masking. It is interesting to investigate whether this is true for any scenario.

## 7 Ethics Statement

This paper proposes a novel task-incremental learning model that can achieve both forgetting prevention and knowledge transfer for a mixed sequence of heterogeneous tasks. We do not foresee any negative consequences for any individual as a result of this research. The consequence of the failure of the system is that the system makes some incorrect predictions, which, we believe, do not have any ethic implications. All the data and pre-trained models are publicly available and our method does not leverage biases in the data.

## Acknowledgements

The work of Bing Liu was supported in part by four National Science Foundation (NSF) grants (1910424, 1838770, 2225427, and 2229876) and a research contract from KDDI.

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

## A    Additional Details about Adapters

TSS leverages adapters to do masking. An adapter is simply a 2-layer fully connected network with layer normalization and residual connections inserted into each Transformer layer, which adapts the output distribution of the pre-trained Transformer language model (LM) *without* modifying LM's original weights (the original LM is fixed). Figure 2 illustrates the LM with adapters.

## B    Additional Details about the Datasets

Recall that TSS uses five datasets. Here we give their detailed statistics.

(1) **ASC.** ASC is more than the traditional sentiment classification because of the additional input of the aspect and the fact that in the same sentence different aspects can have different opinions. For example, "*The picture is good but the sound is poor*" about a TV expresses a *positive* opinion about the aspect "picture" and a *negative* opinion about the aspect "sound". We show more details about these datasets in Table 3.

(2) **CCD, SUM, DRG and NER.** We give their detailed statistics in Table 4.

## C    Hyperparameters

Unless otherwise stated, the same hyperparameters are used in all experiments. The maximum input length is set to 128 for all datasets except for SUM which uses 1024 due to its longer sequences. AdamW optimizer is used. The learning rate is set to 5e-5 for Transformer (search within {5e-2,5e-3,5e-4,5e-5}), 3e-2 for prompt (search within {3e-1,3e-2,3e-3,3e-4,3e-5}), adapter and classifier. The prompt length is set to 20 (search within {10,20,50,80,100,150}) and adapter bottleneck size is 64, following the original paper (Houlsby et al., 2019). The batch size is set to 32 and the number of training epochs is set to 50 with early stopping. $\tau$ in Eq. 1 is set to 0. For classification tasks, a separate classification head is used for each task in the sequence. For generation tasks, a shared LM head is used for all tasks in a sequence. we further set the number of beams to 4 for beam search and constrain the target length in between 30 to 200. For image-based (EWC, Cuber, DAS, HAT, SupSup, CAT, L2P) and RoBERTa-based (CTR) systems, we adapt them for text classification and generation by replacing their feature extractors with BART. For

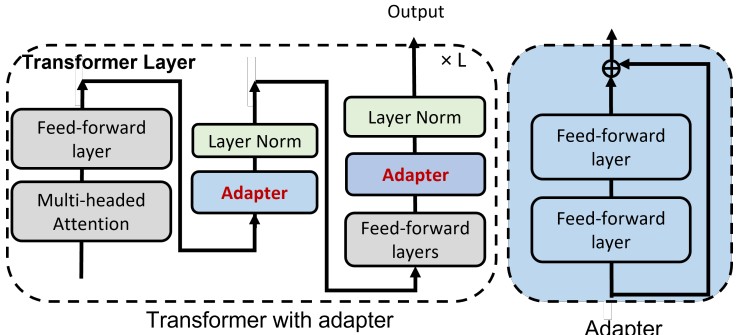

Figure 2: Architecture of Transformer with adapters. An adapter (blue component) is inserted in each layer. Only the blue and green boxes are trainable while the LM is fixed during training.

| Tasks/Domains | #Training | #Validating | #Testing |
|---|---|---|---|
| Speaker | 233 S./352 A./287 P./65 N./0 Ne. | 30 S./44 A./35 P./9 N./0 Ne. | 38 S./44 A./40 P./4 N./0 Ne. |
| Router | 200 S./245 A./142 P./103 N./0 Ne. | 24 S./31 A./19 P./12 N./0 Ne. | 22 S./31 A./24 P./7 N./0 Ne. |
| Computer | 187 S./283 A./218 P./65 N./0 Ne. | 25 S./35 A./23 P./12 N./0 Ne. | 29 S./36 A./29 P./7 N./0 Ne. |
| Nokia6610 | 209 S./271 A./198 P./73 N./0 Ne. | 29 S./34 A./30 P./4 N./0 Ne. | 28 S./34 A./25 P./9 N./0 Ne. |
| Nikon4300 | 131 S./162 A./135 P./27 N./0 Ne. | 15 S./20 A./18 P./2 N./0 Ne. | 15 S./21 A./19 P./2 N./0 Ne. |
| Creative | 582 S./677 A./422 P./255 N./0 Ne. | 68 S./85 A./42 P./43 N./0 Ne. | 70 S./85 A./52 P./33 N./0 Ne. |
| CanonG3 | 190 S./228 A./180 P./48 N./0 Ne. | 25 S./29 A./21 P./8 N./0 Ne. | 24 S./29 A./24 P./5 N./0 Ne. |
| ApexAD | 281 S./343 A./146 P./197 N./0 Ne. | 35 S./43 A./16 P./27 N./0 Ne. | 28 S./43 A./31 P./12 N./0 Ne. |
| CanonD500 | 103 S./118 A./96 P./22 N./0 Ne. | 11 S./15 A./14 P./1 N./0 Ne. | 13 S./15 A./11 P./4 N./0 Ne. |
| Canon100 | 137 S./175 A./123 P./52 N./0 Ne. | 19 S./22 A./20 P./2 N./0 Ne. | 16 S./22 A./21 P./1 N./0 Ne. |
| Diaper | 166 S./191 A./143 P./48 N./0 Ne. | 22 S./24 A./18 P./6 N./0 Ne. | 24 S./24 A./22 P./2 N./0 Ne. |
| Hitachi | 152 S./212 A./153 P./59 N./0 Ne. | 23 S./26 A./19 P./7 N./0 Ne. | 23 S./27 A./14 P./13 N./0 Ne. |
| Ipod | 124 S./153 A./101 P./52 N./0 Ne. | 18 S./19 A./14 P./5 N./0 Ne. | 19 S./20 A./15 P./5 N./0 Ne. |
| Linksys | 152 S./176 A./128 P./48 N./0 Ne. | 19 S./22 A./13 P./9 N./0 Ne. | 20 S./23 A./16 P./7 N./0 Ne. |
| MicroMP3 | 384 S./484 A./340 P./144 N./0 Ne. | 42 S./61 A./48 P./13 N./0 Ne. | 51 S./61 A./39 P./22 N./0 Ne. |
| Nokia6600 | 298 S./362 A./244 P./118 N./0 Ne. | 26 S./45 A./32 P./13 N./0 Ne. | 39 S./46 A./30 P./16 N./0 Ne. |
| Norton | 168 S./194 A./54 P./140 N./0 Ne. | 17 S./24 A./15 P./9 N./0 Ne. | 24 S./25 A./5 P./20 N./0 Ne. |
| Restaurant | 1893 S./3452 A./2094 P./779 N./579 Ne. | 84 S./150 A./70 P./26 N./54 Ne. | 600 S./1120 A./728 P./196 N./196 Ne. |
| Laptop | 1360 S./2163 A./930 P./800 N./433 Ne. | 98 S./150 A./57 P./66 N./27 Ne. | 411 S./638 A./341 P./128 N./169 Ne. |

Table 3: Statistics of the ASC tasks. **S.**: number of sentences; **A**: number of aspects; **P., N., and Ne.**: number aspects with positive, negative and neutral opinions, respectively. Note that the "Restaurant" and "Laptop" have 3 classes of opinion polarities (positive, negative and neutral) while the others have only 2 classes (positive and negative).

| Dataset | Tasks/Domains | #Training | #Validating | #Testing | #Classes |
|---|---|---|---|---|---|
| CCD | Yahoo | 4500 | 500 | 4840 | 10 |
| | AGnews | 1785 | 199 | 1756 | 2 |
| | Amazon | 898 | 100 | 998 | 2 |
| | Dbpedia | 6237 | 693 | 6748 | 14 |
| | Yelp | 900 | 100 | 984 | 4 |
| SUM | icsi | 43 | 10 | 6 | — |
| | ami | 97 | 20 | 20 | — |
| | reddit | 201 | 50 | 250 | — |
| | stack | 205 | 50 | 250 | — |
| | nyt | 200 | 50 | 250 | — |
| | emails | 215 | 50 | 250 | — |
| DRG | taxi | 406 | 71 | 56 | — |
| | hotel | 3366 | 143 | 177 | — |
| | attraction | 298 | 27 | 28 | — |
| | train | 1954 | 196 | 262 | — |
| | restaurant | 569 | 63 | 59 | — |
| NER | conll2003 | 200 | 3250 | 3453 | 9 |
| | wikigold | 200 | 170 | 170 | 9 |
| | btc | 200 | 934 | 934 | 9 |
| | re3d | 200 | 77 | 200 | 21 |
| | gum | 200 | 250 | 1000 | 23 |

Table 4: Statistics of the CCD, SUM, DRE and NER datasets. Number of classes is not applicable to SUM and DRG because they are generation datasets.

LAMOL, we directly run the author provided code. Except for the aforementioned hyper-parameters, all baseline-specific hyper-parameters follow those in their original papers.

## D    Details of CL Baselines

**5 adapter-based methods** (CTR (Ke et al., 2021a), HAT (Serrà et al., 2018), SupSup (Wortsman et al., 2020), CAT (Ke et al., 2020) and CUBER (Lin et al., 2022a)). They all train a shared adapter while SupSup trains popup scores to indicate sub-network on the fixed adapter. HAT is one of the most effective TIL methods with little forgetting. CTR encourages transfer via capsule networks and transfer routing. SupSup uses the similar sub-network discovery method as TSS but cannot do KT at all. CAT and CTR are two systems that deal with both CF and KT. CUBER also deals with CF and KT, but a hard threshold is needed.

**1 prompt-based method** (L2P (Wang et al., 2022)), which trains a prompt pool to transfer task knowledge and a key-value pair prompt selection strategy to select the task-specific prompt (it thus deals with both KT and CF).

**3 baselines that modify the Transformer** (LAMOL (Sun et al., 2020), EWC (Kirkpatrick et al., 2016) and **DAS** (Ke et al., 2023)). LAMOL is a pseudo-replay method using GPT-2. EWC is a regularization method. DAS is a soft-masking method aims to achieve both forgetting prevention and knowledge transfer for continual pre-training. Unlike TSS, it does not discover any sub-network and thus still suffers from forgetting.

## E    Detailed Results for Each System

In Tables 1 and 2 in the main paper, we only report the results of the **main** metrics (MF1 in ASC and CCD, F1 in NER, R1 in SUM and BLEU in DRG). In this section, we give all the results of all the other metrics. We report the detailed results in Tables 5 and 6, we can see TSS is effective in all other metrics.

## F    Difference between our forgetting rate and the one in (Mehta et al., 2021)

Unlike our forgetting rate in Sec. 4, the forgetting rate in (Mehta et al., 2021) is defined as $F'_t = \frac{1}{t-1} \sum_{\tau=1}^{t-1} \max_{\tau' \in \{1,...,t-1\}} (S_{\tau',\tau} - S_{t,\tau})$. Since both metrics measure everything from the standing point of the end of continual learning, i.e., after all tasks are learned, we believe our measure

is more reasonable. Let us use some examples to illustrate,

- (1). if task 1 archives the accuracy 0.5 right after its training, it achieves 0.4 after task 2, and it achieves 0.3 after task 3 (final task). In this case, both measures give the same forgetting 0.2.

- (2). if task 1 archives the accuracy 0.5 right after its training, it achieves 0.8 after task 2, and it achieves 0.82 after task 3 (final task). If we take max $(F'_t)$, then the forgetting is -0.02, but our method will give -0.32. In this case, our method is more reasonable because it precisely shows how much backward transfer (negative value here means backward transfer) has been achieved for task 1 after task 2 and task 3 are learned since both measures evaluate from the same reference point, i.e., after all tasks are learned (the last task is t in both measures, i.e., task 3 in this example).

- (3). If task 1 archives the accuracy 0.5 right after its training, 0.8 after task 2 and 0.4 after task 3 (final task), our metric will give the forgetting 0.1. $F'_t$ will give 0.4. This case is more debatable because $F'_t$ catches the worst forgetting. But since we evaluate after all tasks are learned (the reference point is when the last task is learned), again we believe that our method is more reasonable.

- (4). If task 1 archives the accuracy 0.5 right after its training, 0.1 after task 2 and 0.4 after task 3 (final task), both metrics will give the forgetting 0.1. In this case, $F'_t$ does not catch the worst forgetting of 0.4 (0.5-0.1) in the process. Again, if we agree that we evaluate from the reference point of when the last task is learned, then both measures are fine in this case.

In summary, while we believe ours is more reasonable, a better metric may be designed in the future to characterize forgetting and knowledge transfer in the continual learning process.

## G    Detailed Forgetting Rate for Each Dataset

We report the average forgetting rate of each system in Tables 1 and 2 (in the main paper). In this section, we report the detailed forgetting rate of each system corresponding to each of the above tables.

| Dataset | Similar | | | Dissimilar | | | | | | Average | Average |
| | ASC | | NER | SUM | | | CCD | | DRG | | |
| Model | MF1 | Acc | F1 | R1 | R2 | RL | MF1 | Acc | BLEU | Main | FR |
|---|---|---|---|---|---|---|---|---|---|---|---|
| **Non-continual learning** | | | | | | | | | | | |
| MTL | 92.28 | 94.82 | 63.33 | 39.39 | 10.33 | 35.05 | 90.57 | 90.55 | 25.29 | 62.17 | — |
| MTL (Adapter) | 92.17 | 94.65 | 60.61 | 38.84 | 11.38 | 34.81 | 91.09 | 91.14 | 24.50 | 61.44 | — |
| ONE | 85.55 | 91.50 | 59.33 | 39.07 | 10.71 | 35.25 | 91.07 | 91.09 | 24.14 | 59.83 | — |
| ONE (Adapter) | 83.95 | 90.90 | 56.69 | 38.90 | 11.54 | 35.23 | 90.78 | 90.81 | 23.42 | 58.75 | — |
| ONE (Prompt) | 76.46 | 85.54 | 45.90 | 30.67 | 7.23 | 27.53 | 86.23 | 86.28 | 12.67 | 50.39 | — |
| **Continual learning of heterogeneous Tasks** | | | | | | | | | | | |
| NCL | 88.05 | 92.60 | 34.56 | 22.86 | 4.67 | 20.61 | 81.86 | 82.70 | 8.68 | 47.20 | 13.94 |
| EWC | 88.73 | 92.86 | 28.75 | 18.65 | 21.22 | 16.36 | 83.24 | 83.92 | 7.99 | 45.47 | 13.43 |
| HAT | 87.45 | 92.77 | 50.16 | 35.21 | 9.96 | 31.83 | 89.85 | 90.03 | 20.98 | 56.73 | 0.97 |
| DAS | 90.62 | 93.73 | 43.38 | 25.45 | 5.88 | 23.08 | 82.66 | 84.39 | 15.21 | 51.47 | 10.49 |
| SupSup | 85.83 | 92.41 | 58.93 | 38.23 | 11.44 | 34.88 | 90.66 | 90.68 | 24.71 | 59.67 | 0.00 |
| CAT | 45.79 | 50.08 | 35.24 | 17.44 | 2.43 | 15.50 | 36.62 | 39.01 | 10.21 | 29.06 | 30.27 |
| CTR | 89.37 | 93.28 | 50.16 | 33.98 | 9.97 | 30.80 | 89.96 | 89.98 | 17.63 | 56.22 | 0.95 |
| CUBER | 90.58 | 93.72 | 46.67 | 30.94 | 7.30 | 27.92 | 81.54 | 81.41 | 15.68 | 53.08 | 7.52 |
| L2P | 78.10 | 87.97 | 39.03 | 28.34 | 5.93 | 25.44 | 72.35 | 76.99 | 4.55 | 44.47 | 6.44 |
| TSS | **90.61** | **94.47** | **62.13** | **38.29** | **11.77** | **35.26** | **90.70** | **90.70** | **24.56** | **61.26** | **0.00** |
| **Continual learning of homogeneous Tasks** | | | | | | | | | | | |
| CL | 89.25 | 93.04 | 49.19 | 32.68 | 6.84 | 29.19 | 85.08 | 85.10 | 22.31 | 55.70 | 6.42 |
| EWC | 88.36 | 92.60 | 51.76 | 32.64 | 7.12 | 29.00 | 87.27 | 87.37 | 18.30 | 55.66 | 5.53 |
| HAT | 89.33 | 93.28 | 52.31 | 37.11 | 10.40 | 33.53 | 90.21 | 90.23 | 21.47 | 58.08 | 0.49 |
| DAS | 90.94 | 94.20 | 42.12 | 31.31 | 7.61 | 28.79 | 87.97 | 88.12 | 22.25 | 54.92 | 9.26 |
| SupSup | 85.83 | 92.41 | 58.93 | 38.23 | 11.44 | 34.88 | 90.66 | 90.68 | 24.71 | 59.67 | 0.00 |
| LAMOL | 84.62 | 90.17 | — | 10.88 | 1.39 | 6.87 | 54.44 | 67.04 | 19.96 | — | — |
| CAT | 84.31 | 88.98 | 50.73 | 37.24 | 10.53 | 33.77 | 90.82 | 90.90 | 21.72 | 56.96 | 1.37 |
| CTR | 88.86 | 92.94 | 51.85 | 37.34 | 10.73 | 33.69 | 90.54 | 90.58 | 21.39 | 58.00 | 0.65 |
| CUBER | 91.25 | 94.33 | 47.19 | 30.44 | 6.75 | 27.51 | 89.19 | 89.27 | 21.66 | 55.95 | 5.82 |
| L2P | 74.81 | 84.64 | 44.22 | 26.65 | 4.82 | 23.90 | 85.35 | 85.54 | 8.52 | 47.91 | 2.58 |
| TSS | **91.28** | **94.06** | **63.96** | **38.39** | **11.60** | **35.22** | **90.89** | **90.92** | **24.75** | **61.85** | **0.00** |

Table 5: Performance for **heterogeneous** (40 tasks in total) and **homogeneous** tasks, averaged over 5 random sequences (the **standard deviation** is reported in Table 9). "—" means not applicable. We bold the best performance within CL baselines.

| Dataset | Similar | | | Dissimilar | | | | | | Average | Average |
| | ASC | | NER | SUM | | | CCD | | DRG | | |
| Model | MF1 | Acc | F1 | R1 | R2 | RL | MF1 | Acc | BLEU | Main | FR |
|---|---|---|---|---|---|---|---|---|---|---|---|
| **Non-continual learning** | | | | | | | | | | | |
| ONE | 85.55 | 91.50 | 59.33 | 39.07 | 10.71 | 35.25 | 91.07 | 91.09 | 24.14 | 59.83 | — |
| **Continual learning of heterogeneous Tasks** | | | | | | | | | | | |
| TSS (w/o SD) | 86.54 | 71.55 | 37.03 | 28.04 | 6.39 | 25.57 | 78.29 | 61.91 | 14.85 | 48.95 | 9.02 |
| TSS (w/o SM) | 85.83 | 92.41 | 58.93 | 38.23 | 11.44 | 34.88 | 90.66 | 90.68 | 24.71 | 59.67 | 0.00 |
| TSS (w/o SM; Naive) | 88.89 | 93.36 | 34.79 | 36.11 | 10.27 | 33.15 | 89.99 | 89.99 | 24.19 | 54.80 | 0.00 |
| TSS | **90.61** | **94.47** | **62.13** | **38.29** | **11.77** | **35.26** | **90.70** | **90.70** | **24.56** | **61.26** | **0.00** |
| **Continual learning of homogeneous Tasks** | | | | | | | | | | | |
| TSS (w/o SD) | 91.06 | 94.24 | 40.47 | 30.42 | 7.30 | 27.83 | 88.34 | 88.40 | 22.77 | 54.61 | 9.61 |
| TSS (w/o SM) | 85.83 | 92.41 | 58.93 | 38.23 | 11.44 | 34.88 | 90.66 | 90.68 | 24.71 | 59.67 | 0.00 |
| TSS (w/o SM; Naive) | 90.38 | 94.25 | 62.42 | 38.08 | 11.52 | 35.05 | 90.63 | 90.74 | 24.59 | 61.22 | 0.00 |
| TSS | **91.28** | **94.06** | **63.96** | **38.39** | **11.60** | **35.22** | **90.89** | **90.92** | **24.75** | **61.85** | **0.00** |

Table 6: Ablation experiment results for **heterogeneous** and **homogeneous** tasks - averages over 5 random sequences (the standard deviation is reported in Table 10).

Table 7 and Table 8 show the detailed forgetting rate for both heterogeneous and homogeneous tasks.

**In heterogeneous tasks**, we can see on average TSS and SupSup have the lowest forgetting rate (TSS outperforms SupSup in final performances due to knowledge transfer). While some baselines (e.g. DAS, CUBER, EWC) have a negative forgetting rate (indicating the new knowledge helps similar old tasks), they suffer from large forgetting in dissimilar tasks because they cannot do well in forgetting.

**In homogeneous tasks**, similar to the heterogeneous tasks, we can see on average TSS and SupSup have the lowest forgetting rate (TSS outperforms SupSup in final performances due to knowledge transfer). We also notice the forgetting is in general less than heterogeneous tasks because the tasks in heterogeneous sequence can be more dissimilar (e.g., from summarization to NER).

# H  Standard Deviations

This section reports the standard deviations of the corresponding results in Tables 1 and 2 (in the main paper) of TSS and the considered baselines over 5 runs with random sequences. We only report the CL baselines since they are related to the task order. We can see the results of TSS is stable. SupSup

| | Similar | | | Dissimilar | | | | | DRG | Average |
|---|---|---|---|---|---|---|---|---|---|---|
| Dataset | ASC | | NER | SUM | | | CCD | | | |
| Model | MF1 | Acc | F1 | R1 | R2 | RL | MF1 | Acc | BLEU | FR |
| | | | | | *Continual learning of heterogeneous Tasks* | | | | | | |
| NCL | 1.89 | 3.41 | 26.89 | 15.79 | 6.16 | 8.77 | 8.65 | 7.35 | 16.47 | 13.94 |
| EWC | -0.16 | 0.57 | 29.29 | 19.50 | 6.70 | 10.33 | 3.06 | 6.53 | 15.46 | 13.43 |
| HAT | -0.98 | 0.11 | 1.54 | 2.21 | 1.20 | 1.49 | 0.62 | 1.12 | 1.44 | 0.97 |
| DAS | -0.44 | 0.01 | 21.66 | 13.48 | 5.65 | 7.32 | 8.02 | 7.86 | 9.73 | 10.49 |
| SupSup | 0.00 | 0.00 | 0.00 | 0.00 | 0.00 | 0.00 | 0.00 | 0.00 | 0.00 | **0.00** |
| CAT | 39.99 | 36.92 | 18.30 | 25.23 | 9.99 | 13.93 | 53.85 | 32.99 | 13.96 | 30.27 |
| CTR | -0.15 | -0.13 | 0.72 | 2.31 | 0.64 | 1.39 | 0.47 | 0.38 | 1.42 | 0.95 |
| CUBER | -1.84 | -1.72 | 12.98 | 11.54 | 5.09 | 6.65 | 1.56 | 1.55 | 12.09 | 7.27 |
| L2P | 2.53 | 0.09 | 4.04 | 5.84 | 2.87 | 3.99 | 12.66 | 11.31 | 7.16 | 6.44 |
| TSS | 0.00 | 0.00 | 0.00 | 0.00 | 0.00 | 0.00 | 0.00 | 0.00 | 0.00 | **0.00** |
| | | | | | *Continual learning of homogeneous Tasks* | | | | | | |
| NCL | -0.44 | 0.31 | 15.49 | 8.37 | 5.54 | 5.90 | 5.67 | 5.64 | 3.01 | 6.42 |
| EWC | 7.81 | 2.78 | 7.53 | 2.58 | 1.68 | 2.04 | 8.94 | 8.17 | 0.78 | 5.53 |
| HAT | -0.88 | -0.36 | 0.79 | 1.10 | 1.22 | 0.78 | -0.07 | -0.08 | 1.48 | 0.49 |
| DAS | -0.81 | -0.77 | 31.88 | 7.87 | 3.89 | 4.98 | 3.73 | 3.56 | 3.61 | 9.26 |
| SupSup | 0.00 | 0.00 | 0.00 | 0.00 | 0.00 | 0.00 | 0.00 | 0.00 | 0.00 | **0.00** |
| LAMOL | -1.88 | -1.52 | — | 3.90 | 1.42 | 2.32 | 32.26 | 27.30 | 2.29 | — |
| CAT | 2.05 | 0.81 | 0.81 | 1.85 | 1.41 | 1.37 | -0.13 | -0.21 | 2.26 | 1.37 |
| CTR | -0.38 | -0.08 | 0.58 | 1.43 | 0.90 | 0.86 | -0.27 | -0.27 | 1.91 | 0.65 |
| CUBER | -0.38 | -0.08 | 0.58 | 1.43 | 0.90 | 0.86 | -0.27 | -0.27 | 1.91 | 0.65 |
| L2P | 0.30 | 0.54 | 0.96 | 7.27 | 3.53 | 4.68 | 0.94 | 0.82 | 3.42 | 2.58 |
| TSS | 0.00 | 0.00 | 0.00 | 0.00 | 0.00 | 0.00 | 0.00 | 0.00 | 0.00 | **0.00** |

Table 7: Forgetting rate for **heterogeneous** (40 tasks in total) and **homogeneous** tasks, averaged over 5 random sequences.

| | Similar | | | Dissimilar | | | | | DRG | Average |
|---|---|---|---|---|---|---|---|---|---|---|
| Dataset | ASC | | NER | SUM | | | CCD | | | |
| Model | MF1 | Acc | F1 | R1 | R2 | RL | MF1 | Acc | BLEU | FR |
| | | | | | *Continual learning of heterogeneous Tasks* | | | | | | |
| TSS (w/o SD) | -0.48 | 0.24 | 20.90 | 8.88 | 4.19 | 4.57 | 8.38 | 8.53 | 7.41 | 9.02 |
| TSS (w/o SM) | 0.00 | 0.00 | 0.00 | 0.00 | 0.00 | 0.00 | 0.00 | 0.00 | 0.00 | **0.00** |
| TSS (w/o SM; Naive) | 0.00 | 0.00 | 0.00 | 0.00 | 0.00 | 0.00 | 0.00 | 0.00 | 0.00 | **0.00** |
| TSS | 0.00 | 0.00 | 0.00 | 0.00 | 0.00 | 0.00 | 0.00 | 0.00 | 0.00 | **0.00** |
| | | | | | *Continual learning of homogeneous Tasks* | | | | | | |
| TSS (w/o SD) | -1.71 | -0.77 | 34.14 | 10.18 | 4.94 | 6.62 | 2.83 | 2.74 | 2.62 | 9.61 |
| TSS (w/o SM) | 0.00 | 0.00 | 0.00 | 0.00 | 0.00 | 0.00 | 0.00 | 0.00 | 0.00 | **0.00** |
| TSS (w/o SM; Naive) | 0.00 | 0.00 | 0.00 | 0.00 | 0.00 | 0.00 | 0.00 | 0.00 | 0.00 | **0.00** |
| TSS | 0.00 | 0.00 | 0.00 | 0.00 | 0.00 | 0.00 | 0.00 | 0.00 | 0.00 | **0.00** |

Table 8: Forgetting rate for ablation of **heterogeneous tasks** (40 tasks in total) and **homogeneous** tasks, averaged over 5 random sequences.

and TSS (w/o SM) discover different sub-networks for different tasks, so they are not related to the task order and are not reported in the table. We report the standard deviations for all metrics for completeness.

**In heterogeneous tasks**, Table 9 and Table 10 report the standard deviations of TSS and the considered baselines over 5 runs with random sequences. We can see the results of TSS and its variants are stable.

**In homogeneous tasks**, similar to heterogeneous tasks, we can see the results of TSS and its variants are stable.

## I Results for Individual Sequences

In Table 1 in the main text, we reported the results averaged over 5 random sequences (different task orders). In this section, we give the results of TSS of each sequence in Table 11. We can see that the order indeed affects the results but not by much. In summary, we believe the average over random sequences in Table 1 (in the main text/paper) can show the effectiveness of TSS.

## J Execution Time and Number of Parameters

Table 12 reports the number of parameters (regardless of trainable or non-trainable), training execution times for different CL models. Our experiments were run on 4 GeForce GTX 2080 Ti with

| | Similar | | | Dissimilar | | | | | |
|---|---|---|---|---|---|---|---|---|---|
| Dataset | ASC | | NER | SUM | | | CCD | | DRG |
| Model | MF1 | Acc | F1 | R1 | R2 | RL | MF1 | Acc | BLEU |
| **Continual learning of heterogeneous tasks** | | | | | | | | | |
| NCL | ±0.0133 | ±0.0043 | ±0.1397 | ±0.1079 | ±0.0251 | ±0.0954 | ±0.0150 | ±0.0044 | ±0.0527 |
| EWC | ±0.0114 | ±0.0050 | ±0.1278 | ±0.1047 | ±0.2167 | ±0.0947 | ±0.0171 | ±0.0155 | ±0.0914 |
| HAT | ±0.0395 | ±0.0152 | ±0.0078 | ±0.0041 | ±0.0024 | ±0.0046 | ±0.0020 | ±0.0023 | ±0.0119 |
| DAS | ±0.0126 | ±0.0072 | ±0.0348 | ±0.1116 | ±0.0300 | ±0.1000 | ±0.0241 | ±0.0375 | ±0.0638 |
| CAT | ±0.1732 | ±0.2137 | ±0.0034 | ±0.0549 | ±0.0318 | ±0.0794 | ±0.1176 | ±0.2163 | ±0.0333 |
| CTR | ±0.0447 | ±0.0023 | ±0.0712 | ±0.0278 | ±0.0058 | ±0.0029 | ±0.1492 | ±0.0385 | ±0.0704 |
| CUBER | ±0.0175 | ±0.0107 | ±0.0589 | ±0.0682 | ±0.0208 | ±0.0598 | ±0.0394 | ±0.0396 | ±0.0631 |
| L2P | ±0.0173 | ±0.0260 | ±0.0225 | ±0.0328 | ±0.0153 | ±0.0298 | ±0.0759 | ±0.0494 | ±0.0287 |
| TSS | ±0.0058 | ±0.0124 | ±0.0503 | ±0.0025 | ±0.0018 | ±0.0021 | ±0.0047 | ±0.0034 | ±0.0058 |
| **Continual learning of homogeneous tasks** | | | | | | | | | |
| NCL | ±0.0040 | ±0.0027 | ±0.0767 | ±0.0060 | ±0.0055 | ±0.0066 | ±0.0233 | ±0.0230 | ±0.0060 |
| EWC | ±0.0093 | ±0.0063 | ±0.0707 | ±0.0132 | ±0.0053 | ±0.0113 | ±0.0227 | ±0.0216 | ±0.0217 |
| HAT | ±0.0130 | ±0.0042 | ±0.0063 | ±0.0130 | ±0.0049 | ±0.0115 | ±0.0028 | ±0.0028 | ±0.0094 |
| DAS | ±0.0075 | ±0.0024 | ±0.0647 | ±0.0091 | ±0.0070 | ±0.0074 | ±0.0178 | ±0.0175 | ±0.0094 |
| LAMOL | ±0.0085 | ±0.0039 | — | ±0.0125 | ±0.0042 | ±0.0052 | ±0.0317 | ±0.0258 | ±0.0068 |
| CAT | ±0.0096 | ±0.0021 | ±0.0106 | ±0.0104 | ±0.0040 | ±0.0090 | ±0.0019 | ±0.0021 | ±0.0051 |
| CTR | ±0.0080 | ±0.0030 | ±0.0049 | ±0.0116 | ±0.0052 | ±0.0096 | ±0.0009 | ±0.0010 | ±0.0127 |
| CUBER | ±0.0072 | ±0.0031 | ±0.0543 | ±0.0157 | ±0.0061 | ±0.0119 | ±0.0069 | ±0.0062 | ±0.0090 |
| L2P | ±0.0650 | ±0.0361 | ±0.0022 | ±0.0215 | ±0.0047 | ±0.0203 | ±0.0181 | ±0.0176 | ±0.0216 |
| TSS | ±0.0031 | ±0.0035 | ±0.0032 | ±0.0021 | ±0.0020 | ±0.0015 | ±0.0024 | ±0.0026 | ±0.0035 |

Table 9: Standard deviations of the corresponding metrics of the proposed TSS model and the baselines on the **heterogeneous** and **homogeneous** tasks.

| | Similar | | | Dissimilar | | | | | |
|---|---|---|---|---|---|---|---|---|---|
| Dataset | ASC | | NER | SUM | | | CCD | | DRG |
| Model | MF1 | Acc | F1 | R1 | R2 | RL | MF1 | Acc | BLEU |
| **Continual learning of heterogeneous tasks** | | | | | | | | | |
| TSS (w/o SD) | ±0.0866 | ±0.3657 | ±0.1328 | ±0.0536 | ±0.0216 | ±0.0473 | ±0.0942 | ±0.3222 | ±0.0541 |
| TSS (w/o SM; Naive) | ±0.0096 | ±0.0063 | ±0.0307 | ±0.0106 | ±0.0063 | ±0.0107 | ±0.0028 | ±0.0029 | ±0.0049 |
| TSS | ±0.0058 | ±0.0124 | ±0.0503 | ±0.0025 | ±0.0018 | ±0.0021 | ±0.0047 | ±0.0034 | ±0.0058 |
| **Continual learning of homogeneous tasks** | | | | | | | | | |
| TSS (w/o SD) | ±0.0107 | ±0.0066 | ±0.0695 | ±0.0088 | ±0.0056 | ±0.0071 | ±0.0274 | ±0.0268 | ±0.0039 |
| TSS (w/o SM; Naive) | ±0.0102 | ±0.0040 | ±0.0067 | ±0.0024 | ±0.0014 | ±0.0023 | ±0.0019 | ±0.0016 | ±0.0019 |
| TSS | ±0.0031 | ±0.0035 | ±0.0032 | ±0.0021 | ±0.0020 | ±0.0015 | ±0.0024 | ±0.0026 | ±0.0035 |

Table 10: Standard deviations of the corresponding metrics of the proposed TSS model and the ablation experiments on the **heterogeneous** and **homogeneous** tasks.

| | | Similar | | | Dissimilar | | | | | | Average |
|---|---|---|---|---|---|---|---|---|---|---|---|
| Task | Order | ASC | | NER | SUM | | | CCD | | DRG | |
| | | MF1 | Acc | F1 | R1 | R2 | RL | MF1 | Acc | BLEU | Main |
| Heterogeneous | 0 | 90.61 | 94.47 | 62.13 | 38.29 | 11.77 | 35.26 | 90.70 | 90.70 | 24.56 | 61.26 |
| | 1 | 90.07 | 93.66 | 62.00 | 38.56 | 11.65 | 35.35 | 90.21 | 90.23 | 24.25 | 61.02 |
| | 2 | 89.34 | 93.46 | 48.83 | 37.80 | 11.33 | 34.75 | 91.35 | 91.20 | 25.92 | 58.65 |
| | 3 | 89.59 | 93.65 | 58.53 | 38.39 | 11.78 | 35.24 | 90.24 | 90.81 | 25.25 | 60.40 |
| | 4 | 88.95 | 90.82 | 61.25 | 38.31 | 11.45 | 35.09 | 90.07 | 90.38 | 25.07 | 60.73 |
| Homogeneous | 0 | 91.57 | 94.40 | 63.96 | 38.46 | 11.38 | 35.19 | 91.02 | 91.08 | 24.89 | 61.98 |
| | 1 | 91.65 | 94.45 | 63.19 | 38.53 | 11.62 | 35.28 | 90.84 | 90.86 | 24.07 | 61.66 |
| | 2 | 90.96 | 93.62 | 63.26 | 38.58 | 11.85 | 35.40 | 91.28 | 91.34 | 24.81 | 61.78 |
| | 3 | 91.48 | 94.39 | 63.01 | 38.14 | 11.72 | 35.06 | 90.65 | 90.65 | 25.02 | 61.66 |
| | 4 | 90.94 | 93.78 | 63.37 | 38.06 | 11.33 | 34.97 | 90.65 | 90.67 | 24.97 | 61.60 |

Table 11: Results for TSS in different orders for **heterogeneous and homogeneous** tasks.

48G GPUs memory. We only report the training time for **heterogenous** tasks because the training time is related to datasets and one of the main goal of TSS is to work well in the challenging **heterogeneous** scenario. we can see TSS is very efficient, ranking among the least number of parameters and shortest training time among all the baselines considered.

**Saved binary gates for each task.** For each task, the memory usage for the saved binary gates is on average 3.1M bytes, which is 7 times less than saving all the information of the adapter (25.2M), not to mention if one saves the whole LM (3.6G). The number of accumulated importance values is constant, which has the same size as the adapter.

| Model | #Params (M) | Training time (min) |
|-------|-------------|---------------------|
| NCL | 896.9 | 600 |
| EWC | 896.9 | 600 |
| HAT | 958.0 | 600 |
| DAS | 896.9 | 600 |
| LAMOL | 124.4 | 550 |
| CAT | 896.9 | 24,000 |
| CTR | 1,536 | 48,000 |
| CUBER | 896.9 | 600 |
| L2P | 899.0 | 650 |
| TSS | 896.9 | 600 |

Table 12: Network size (#parameters in millions, regardless of trainable or non-trainable) and average training time per task of each CL model measured in minutes for **heterogenous** tasks. Here we report #parameters without including the memory buffer.