# OpenReview forum: "Sub-network Discovery and Soft-masking for Continual Learning of Mixed Tasks"
_EMNLP/2023/Conference — EMNLP 2023 Findings_

### Official Review · Reviewer_jmje · 2023-07-20

**Typos Grammar Style And Presentation Improvements:** None
**Soundness:** 3

**Excitement:**

2: Mediocre: This paper makes marginal contributions (vs non-contemporaneous work), so I would rather not see it in the conference.

**Missing References:**

[1] Ermis B, Zappella G, Wistuba M, et al. Memory efficient continual learning with transformers[J]. Advances in Neural Information Processing Systems, 2022, 35: 10629-10642.

[2] Zheng J, Liang Z, Chen H, et al. Distilling Causal Effect from Miscellaneous Other-Class for Continual Named Entity Recognition[C]//Proceedings of the 2022 Conference on Empirical Methods in Natural Language Processing. 2022: 3602-3615.

[3] Xia Y, Wang Q, Lyu Y, et al. Learn and review: Enhancing continual named entity recognition via reviewing synthetic samples[C]//Findings of the Association for Computational Linguistics: ACL 2022. 2022: 2291-2300.

**Paper Topic And Main Contributions:**

This paper focus on the continually learning a mixed sequence. They propose two subnetwork discovery with soft-masking to allevaite catastrophic forgetting and importance computation to learn a mixed sequence. They also conduct extensive experiments on different NLP tasks and compare with many methods.

**Questions For The Authors:**

Question A: The paper states that "To our knowledge, only one method has been proposed to learn a sequence of mixed tasks.". Why other task-incremental learning methods are not suitable for learning a sequence of mixed tasks? What are the special designs in your method enable learning a sequence of mixed tasks?
Question B: Why the proposed method is built upon adapter? Is it an empirical choice?

**Reasons To Accept:**

1. The extensive experiments show the effectiveness of the proposed method under the setting of learning mixed sequence continually.

**Reasons To Reject:**

1. The significance of studying continual learning of a mixed sequence of tasks is unclear. This paper does not provide a formal definition of  "continual learning of a mixed sequence" and clearify what is the exact meaning of "similar tasks" and "disimilar tasks". In a practical scenario of task-incremental learning, the model should not have assumptions about the tasks to be learned. Therefore, I think the setting is strange and unrealistic.
2. The core idea of the proposed method (e.g., subnetwork discovery and importance score computation) is not innovative. Furthermore, there are no strong connections between the proposed method and the setting of continual learning of a mixed sequence of tasks. It seems that the proposed method can be apply to conventional task-incremental learning settings.
3. The method part of the paper is hard to follow (e.g., Figure 1 is difficult to understand). Besides, the rationale of some model design is quite confusing.
4. The paper claim that the proposed method encourage knowlege transfer, but the results of some key metrics (e.g., forward/backward transfer) are not shown in the experiments.

**Reproducibility:**

4: Could mostly reproduce the results, but there may be some variation because of sample variance or minor variations in their interpretation of the protocol or method.

**Reviewer Confidence:**

4: Quite sure. I tried to check the important points carefully. It's unlikely, though conceivable, that I missed something that should affect my ratings.

---

> ### Author Rebuttal · Authors · 2023-08-29
>
> > Q1: The significance of studying continual learning of a mixed sequence of tasks is unclear
>
> The concepts of “mixed sequence”, “similar” and “dissimilar tasks” have been used in the literature [1,2]. “mixed sequence” is a very practical and general scenario as we cannot guarantee that all tasks are similar or dissimilar in a continual learning application. It is very true that continual learning should not assume the types of tasks to be learned. Precisely because of this, we follow [1,2] to create similar and dissimilar tasks and mix them to create complex task sequences. We defined the similar tasks as tasks sharing the same or similar labels from different domains (see the details in L428-429). In fact, many existing continual learning works implicitly assume tasks are dissimilar and focus on forgetting prevention only. In this work, we do not have such an assumption. Instead our tasks in a sequence can be similar, dissimilar or a mixture of them, which is the most general case.  We also allow the tasks to be homogeneous (same type of tasks) or heterogeneous (different types of tasks). As a result, we believe that our settings are much more general and realistic than the existing work that assumes tasks are dissimilar and homogeneous.
>
> [1]: Continual learning of a mixed sequence of similar and dissimilar tasks, Ke et al., NeurIPS 2020
> [2]: Continual Sequence Generation with Adaptive Compositional Modules, Zhang et al., ACL 2022
>
> > Q2: The core idea of the proposed method (e.g., subnetwork discovery and importance score computation) is not innovative.
>
> We are eager to address your concerns. We hope you can give us more details and also prior work references that are similar to our work.
>
> TSS addresses both forgetting prevention and knowledge transfer in a wide range of homogeneous and heterogeneous tasks. Only CAT [1] tries to do the same but its similarity detection can go very wrong and TSS does not need such task similarity detection.
>
> > Q3: Furthermore, there are no strong connections between the proposed method and the setting of continual learning of a mixed sequence of tasks. It seems that the proposed method can be apply to conventional task-incremental learning settings.
>
> The conventional TIL setting uses dissimilar and homogeneous tasks. Yes, TSS can work in this setting. We can see from Table 2 that TSS works very well. What’s more, TSS also works well in the other much more challenging settings (i.e., heterogeneous, similar tasks, and mixed sequences). TSS can achieve these because our soft-masking mechanism can transfer knowledge and sub-network discovery can prevent forgetting, which are the two key aspects needed in the general setting beyond the conventional TIL.
>
> > Q4: The method part of the paper is hard to follow (e.g., Figure 1 is difficult to understand). Besides, the rationale of some model design is quite confusing.
>
> We will improve our writing to make everything clearer. We noticed that reviewers dnuC and Naqz thought our paper was easy to understand.
>
> About Figure 1, we use one adapter layer as an example to show the two functions of TSS. For sub-network discovery (left-hand side), we train the popup scores ($s_l^{(t)}$) with soft-masking. The forward propagation on the top left shows how the popup scores are used in the computation graph and the backward propagation on the bottom left shows how we use importance to soft-mask the gradients. For importance computation (right-hand side), the forward propagation is exactly the same as the one in sub-network discovery but we use the gradients of the popup scores to indicate the importance (bottom right) instead of updating the popup scores as in sub-network discovery.
>
> > Q5: The paper claim that the proposed method encourage knowlege transfer, but the results of some key metrics (e.g., forward/backward transfer) are not shown in the experiments.
>
> We reported forgetting rate (L535), which is essentially a backward transfer metric. If it is negative, there is backward transfer; if it is positive, there is forgetting. In Table 1, we can see that none of the systems has backward transfer and most baselines systems have forgetting except SupSup. Our TSS performs better than all baselines including SupSup.
>
> As for the forward transfer, we use ONE as control (L544-L551) and observe that TSS can do better in similar tasks which indicates positive forward transfer. We can also report forward performance as in [1], i.e., comparing the performance of ONE and the performance of the proposed method when it learns each task. To strengthen our paper, below we report the forward results as in [1] for the popular baseline NCL as well as the top performing baselines CTR and HAT (average over 5 random sequences). Note that SupSup trains individual tasks independently, so it has no forward transfer.
>
> Heterogeneous tasks
> |  Model  | ASC       | NER       | SUM       | CCD       | DRG       |           |
> |:-------:|-----------|-----------|-----------|-----------|-----------|-----------|
> |         |    MF1    |     F1    |     R1    |    MF1    |    BLEU   |  Average  |
> | NCL     | 89.14     | 61.80     | 38.36     | 89.78     | 24.96     | 60.81     |
> | HAT     | 88.33     | 51.47     | 37.77     | 90.87     | 23.28     | 58.35     |
> | CTR     | 89.22     | 50.88     | 36.29     | 90.44     | 19.05     | 57.18     |
> | **TSS** | **90.61** | **62.13** | **38.39** | **90.90** | **24.98** | **61.40** |
>
> Homogenous tasks
> | Model | ASC       | NER       | SUM       | CCD       | DRG       |           |
> |:-----:|-----------|-----------|-----------|-----------|-----------|-----------|
> |       |    MF1    |     F1    |     R1    |    MF1    |    BLEU   |  Average  |
> | NCL   | 88.84     | 63.96     | 38.98     | 90.39     | 24.69     | 61.37     |
> | HAT   | 88.56     | 52.89     | 38.58     | 90.04     | 22.79     | 58.57     |
> | CTR   | 88.45     | 52.39     | 38.86     | 90.31     | 23.18     | 58.64     |
> | TSS   | **91.28** | **63.96** | **38.99** | **90.89** | **24.75** | **61.97** |
>
> We can see the forward transfer performance on both heterogenous and homogenous tasks which again show that TSS has strong forward knowledge transfer.
>
> > Q6: The paper states that "To our knowledge, only one method has been proposed to learn a sequence of mixed tasks.". Why other task-incremental learning methods are not suitable for learning a sequence of mixed tasks? What are the special designs in your method enable learning a sequence of mixed tasks?
>
> The reason is that most existing task-incremental learning (TIL) methods implicitly assume tasks to be dissimilar and homogeneous. As a result, they only focus on forgetting prevention. While some also do knowledge transfer, they do not consider the heterogeneous tasks and mixed sequences of similar and dissimilar tasks, which are significantly more challenging. Thus, they do not have mechanisms to automatically balance knowledge transfer and forgetting prevention when both are needed in a single sequence. In this work, we argue that the key to learning a mixed sequence of tasks is that the system should not have a priori (L109) information about what knowledge can be transferred and what knowledge should only be learned from the current task data. One way to address this is to detect task similarity as CAT [1], but the detection could be wrong, which causes serious forgetting. TSS proposes to maintain information from previous tasks via soft-masks and use all the information as initialization (L361-L367) to let the current task automatically choose the previous knowledge to use or to transfer. This naturally gets rid of task similarity detection and achieves both forgetting prevention and knowledge transfer in a mixed sequence. We can also see from the experimental results that our method outperforms strong baselines.
>
> > Q7: Why the proposed method is built upon adapter? Is it an empirical choice?
>
> As a parameter-efficient tuning method, the adapter is widely used, including in continual learning [3,4,5]. But we believe our method is not limited to adapters. Other parameter-efficient tuning methods and even the conventional fine-tuning can be used too, as the sub-network discovery and soft-masking are model-agnostic by design.
>
> [3]: Achieving forgetting prevention and knowledge transfer in continual learning, Ke et al., NeurIPS 2021
> [4]: Adapting BERT for continual learning of a sequence of aspect sentiment classification tasks, Ke et al., NAACL 2021
> [5]: Continual Sequence Generation with Adaptive Compositional Modules, Zhang et al., ACL 2022
>
> > Q8: Missing references
> [R1] Ermis B, Zappella G, Wistuba M, et al. Memory efficient continual learning with transformers[J]. Advances in Neural Information Processing Systems, 2022, 35: 10629-10642.
> [R2] Zheng J, Liang Z, Chen H, et al. Distilling Causal Effect from Miscellaneous Other-Class for Continual Named Entity Recognition[C]//Proceedings of the 2022 Conference on Empirical Methods in Natural Language Processing. 2022: 3602-3615.
> [R3] Xia Y, Wang Q, Lyu Y, et al. Learn and review: Enhancing continual named entity recognition via reviewing synthetic samples[C]//Findings of the Association for Computational Linguistics: ACL 2022. 2022: 2291-2300.
>
> Thanks for pointing out these related works. [R1] (ADA) extends conventional adapters by proposing a method to distill knowledge between adapters in continual learning.  It keeps a fixed stack of K adapters, and when a new Adapter needs to be added to the stack, ADA takes the adapter that has the highest similarity with the new task and distills its knowledge to the new task. This is very different from TSS as TSS uses only a single adapter but discovers different sub-networks. In ADA, while distillation can help mitigate forgetting, it is insufficient. Its similarity detection could also be wrong like CAT and cause forgetting. TSS does not have these problems because it discovers different sub-networks and does not do similarity comparison. [R2] and [R3] are both about entity-type incremental learning which is a kind of class-incremental learning (CIL). We focus on task-incremental learning (TIL), which is a different setting than CIL as stated in footnote 2. We will include a discussion about these papers in our revision.

---

### Official Review · Reviewer_Naqz · 2023-07-29

**Typos Grammar Style And Presentation Improvements:** 1) In Equation (3), the subscripts on…
**Soundness:** 4

**Excitement:**

3: Ambivalent: It has merits (e.g., it reports state-of-the-art results, the idea is nice), but there are key weaknesses (e.g., it describes incremental work), and it can significantly benefit from another round of revision. However, I won't object to accepting it if my co-reviewers champion it.

**Missing References:**

1) Konishi, T., Kurokawa, M., Ono, C., Ke, Z., Kim, G., & Liu, B. (2023). Parameter-Level Soft-Masking for Continual Learning. arXiv preprint arXiv:2306.14775.

**Paper Topic And Main Contributions:**

This paper proposed a task incremental learning method (TSS) to achieve both forgetting prevention and knowledge transfer on a range of NLP problems. Specifically, TSS finds a sub-network for each task to avoid catastrophic forgetting and soft-mask the gradient to encourage knowledge transfer.

**Questions For The Authors:**

1) Why were small training samples (less than 30,000) used from 40 different tasks in the experiments? Did the authors attempt to use all the training samples for tasks in a smaller homogeneous sequence for a fair comparison of TSS against other continual learning methods? Was the relationship between the proposed method's performance and the number of training samples analyzed?

2) Was the method compared with a replay-based baseline given the small size of the training dataset (less than 30,000 samples)?

3) Did the authors analyze the relationship between popup scores of different tasks? Are popup scores closer for similar tasks? How does the proposed initialization and soft gradient masking technique affect the learned popup scores?

4) In Section 3.1.1, why was there no specific explanation for the chosen value of the threshold parameter (ε)? Is this parameter set uniformly for all tasks, or do different tasks have different settings?

5) In Section 3.1.2, what is the calculation method for normalizing and rounding the gradient-based importance using Tanh()? Why do the importance scores after Tanh() activation fall within the [0, 1] range, despite the original range being [-1, 1]? What impact does normalizing the importance values to -μ for parameters that initially had an importance of 0 have on the subsequent importance calculations?

6) In Section 4.2, the authors reported average performance on all datasets. However, since different tasks have different evaluation metrics, why did they choose to average F1, BLEU, and ROUGE-1 scores? Why not normalize task performances before averaging?

7) In Section 4.2, why did the authors use ONE as a control for analyzing Knowledge Transfer (KT) when ONE and TSS have different model architectures? Wouldn't it be more reasonable to compare each task's performance independently with its performance in the task sequence?

**Reasons To Accept:**

- The paper is well-structured and easy to understand.
- Introducing a parameter importance-based gradient soft-mask on top of the SupSup method has enhanced the method's knowledge transfer capability.
- Empirical study shows that the proposed method is effective and outperforms CL baselines.

**Reasons To Reject:**

The experiment setting raises concerns regarding using small training set sizes for the 40 task datasets involved in the experiments. This approach gives rise to several issues that should be addressed:

1) **Questionable Applicability**: Continual learning methods are commonly applied in scenarios with large training datasets, where the storage of data and training of new models become crucial considerations. However, the experimental setup in the paper employs less than 30k training data, significantly deviating from real-world application scenarios. This poses doubts about the applicability of the findings to practical situations.
2) **Impact on CL Baselines**: Many CL baselines rely on using the training set of the current task to train adapters or perform fine-tuning on pre-trained models. With the training sets being drastically reduced in size, these methods may suffer from insufficient data, resulting in weakened performance.
3) **Missing Replay-based Baseline**: The paper fails to include a replay-based method as a baseline, which is a challenging approach to beat and should be considered for a comprehensive evaluation.
4) **Ignoring Resource Disparities**: Real-world applications often involve tasks with varying resource availability, directly impacting their relative difficulty. By limiting the training data size uniformly across all tasks, the paper fails to account for the resource differences among them, potentially obscuring important insights.

In conclusion, the experiment setting requires revision to address the aforementioned issues and enhance the validity and applicability of the study's conclusions. Consideration should be given to utilizing more substantial training datasets that align better with real-world scenarios, ensuring the evaluation of CL baselines is based on sufficient data, including the replay-based method as part of the baseline comparisons, and acknowledging and accounting for the resource disparities among different tasks.

**Reproducibility:**

3: Could reproduce the results with some difficulty. The settings of parameters are underspecified or subjectively determined; the training/evaluation data are not widely available.

**Reviewer Confidence:**

4: Quite sure. I tried to check the important points carefully. It's unlikely, though conceivable, that I missed something that should affect my ratings.

---

> ### Author Rebuttal · Authors · 2023-08-29
>
> > Q1: Questionable Applicability. Continual learning are commonly applied in scenarios with large training datasets...
>
> There may be a misunderstanding here. Continual learning research does not have any requirement or restriction on the dataset size. If we think about human learning, we learn continually all the time and we almost always learn in a few-shot manner rather than collect a large amount of data before learning.
>
> For the datasets we used in the paper, they have been used extensively in the literature, e.g., ASC has been used in [1,2], CCD has been used in [3,4], NER and SUM have been used in [5], and DRG has been used in [6]. Except for NER and CCD, where we do random sampling due to our compute resource limitations, we directly use the original full datasets (ASC, SUM, DRG). We regard TSS’s ability to work well with “small” datasets as an advantage rather than disadvantage since labeling data is very expensive. Nevertheless, we agree that experiments on larger datasets can help strengthen our work.
> Following your suggestion, we ran experiments on the large-CCD [3] (76k samples) and large-NER (25k samples). The latest dataset statistic for large-CCD and large-NER are:
> | Large-CCD   | Tasks/Domains | #Train 	| #Test   | #Class |
> |-----------|---------------|------------|------------|--------|
> | AGNews	| News      	| 8,000  	| 7,600  	| 4  	|
> | Yelp  	| Sentiment 	| 10,000 	| 7,600  	| 5  	|
> | Amazon	| Sentiment 	| 10,000 	| 7,600  	| 5  	|
> | DBPedia   | Wikipedia 	| 28,000 	| 7,600  	| 14 	|
> | Yahoo 	| QA        	| 20,000 	| 7,600  	| 10 	|
> | **Total** | **---**   	| **76,000** | **38,000** | **38** |
>
> | Large-NER   | #Train 	| #Test  | #Class |
> |-----------|------------|-----------|--------|
> | conll2023 | 14,041 	| 3,453 	| 9  	|
> | wikigold  | 1,356  	| 170   	| 9  	|
> | btc   	| 7,470  	| 934   	| 9  	|
> | re3d  	| 688    	| 200   	| 21 	|
> | gum   	| 2,245  	| 1,000 	| 23 	|
> | **Total** | **25,800** | **5,757** | **71** |
>
> Due to the time constraint, we compare with popular baselines NCL and ONE as well as the top competitive baselines SupSup and HAT in homogenous tasks (average Macro-F1 over 5 random sequences).
> |  Model  | Large-NER | FR       | Large-CCD | FR       |
> |:-------:|-----------|----------|-----------|----------|
> | ONE     | 76.68     | ---      | 78.14     | ---      |
> | NCL     | 42.57     | 43.65    | 72.93     | 6.62     |
> | SupSup  | 73.96     | 0.00     | 78.15     | 0.00     |
> | HAT     | 61.87     | 0.36     | 77.05     | 0.36     |
> | **TSS** | **76.84** | **0.00** | **78.38** | **0.00** |
>
> We can observe that TSS is still outperforming other baselines, illustrating the effectiveness of TSS in larger datasets. Unlike the improvement observed in Table 1 on NER, we note that the TSS is similar to ONE in the large-NER. This is because knowledge transfer is less obvious when more data for each task is available.
>
> [1]: Achieving forgetting prevention and knowledge transfer in continual learning, Ke et al., NeurIPS 2021
> [2]: Adapting BERT for continual learning of a sequence of aspect sentiment classification tasks, Ke et al., NAACL 2021
> [3]: Continual Learning for Text Classification with Information Disentanglement Based Regularization, Huang et al., NAACL 2021
> [4]: Episodic memory in lifelong language learning, de Masson d’auttume et al., NeurIPS 2019
> [5]: LFPT5: A unified framework for lifelong few-shot language learning based on prompt tuning of T5, Qin et al., ICLR 2022
> [6]: Continual Learning in Task-Oriented Dialogue Systems, Madotto et al., EMNLP 2021
>
> > Q2: Impact on CL Baselines. Existing methods may suffer from insufficient data...
>
> Related to Q1, we believe that working well on less data is an advantage. We also want to note that we do not reduce the data size for ASC, SUM and DRG. In fact, random sampling is a common preprocessing in continual learning papers to reduce the amount of computation. For example, [3] samples 2000 samples per class for CCD. Following your suggestion, we have conducted additional experiments on larger datasets in the answer to Q1.
>
> > Q3: Missing Replay-based Baseline
>
> We didn’t include replay-based methods as we thought it was unnecessary because we have included multitask learning (MTL) (which replays all the past data) as the upper bound, and TSS is comparable to MTL. Also, LAMOL is a popular pseudo-replay method and TSS outperforms LAMOL by a large margin. We agree that adding replay-based baselines is beneficial. We added a popular replay-based method DER++ [8] in homogenous sequences (50 replay samples are saved for each task, which is the largest memory we can apply). We could not run on heterogeneous task sequences on DER++ due to a huge memory requirement by DER++. DER++ has been used in [1,7] and has been shown to be the best replay-based method [7]. All hyperparameters follow those of [1,7]. Results are averaged over 5 random sequences.
>
> |  Model  | ASC   	| NER   	| SUM   	| CCD   	| DRG   	|	Avg.   | Avg. 	|
> |:-------:|-----------|-----------|-----------|-----------|-----------|:---------:|----------|
> |     	|	MF1	| 	F1	| 	R1	|	MF1	|	BLEU   |	Main   | FR   	|
> | DER++   | 90.88 	| 61.74 	| 36.45 	| 86.51 	| 20.65 	| 59.25 	| 5.11 	|
> | **TSS** | **91.28** | **63.96** | **38.39** | **90.89** | **24.75** | **61.85** | **0.00** |
>
> We can see that TSS outperforms DER++, which further illustrates the effectiveness of TSS.
>
> [7]: Continual Pre-training of Language Models, Ke et al., ICLR 2022
> [8]: Dark Experience for General Continual Learning: a Strong, Simple Baseline, Buzzega et al., NeurIPS 2020
>
> > Q4: Ignoring Resource Disparities. By limiting the training data size uniformly across all tasks...
>
> In fact, most of our datasets are imbalanced except one. Thus, we have a good mix of balanced and imbalanced datasets, which is desirable for good evaluation. ASC is highly imbalanced (see Table 3 in Appendix for statistics) and TSS works very well on this data set. SUM and DRG are imbalanced too. The only balanced dataset is NER (see Table 4 in Appendix for statistics). TSS actually works well on the resource disparities scenario.
>
> > Q5: Why were small training samples (less than 30,000) used from 40 different tasks in the experiments? Did the authors attempt to use all the training samples for tasks in a smaller homogeneous sequence for a fair comparison of TSS against other continual learning methods?
>
> We have added experiments with larger datasets in the answer of Q1. Based on the new results and those in Table 1 in the paper, we can see that TSS outperforms baselines under both “small” and “large” datasets. Along with the added datasets, experiments on SUM, NER, DRG and ASC all use their full datasets. For CCD, to our knowledge, none of the existing continual learning work uses the full dataset [3,4] as it is huge (> 500k training samples). We followed [3] and evaluated our method on the large-CCD datasets as well (see the dataset sizes in our answer to your Q1). For the ”smaller sequence”, we have evaluated our method with sequences of various lengths, ranging from 5 to 40 in the paper, which we believe is desirable for good evaluation.
>
> > Q6: Was the relationship between the proposed method's performance and the number of training samples analyzed?
>
> Comparing the added large-NER with the NER (small) in Table 1, we observe that larger datasets improve the performance of both TSS and the baselines, but TSS continues to outperform baselines. Note that the results for large-CCD and CCD (small) in Table 1 are not comparable because CCD (small)  uses a binary classification version of Amazon and Yelp (2 datasets in CCD) while large-CCD from [3] uses the 5-class classification version of these two datasets.
>
> > Q7: Did the authors analyze the relationship between popup scores of different tasks? Are popup scores closer for similar tasks? How does the proposed initialization and soft gradient masking technique affect the learned popup scores?
>
> Popup scores are used to identify what network parameters are useful for each task. We do not believe there is any relationship between them for different tasks. We also note that TSS does not employ any task similarity detection method and thus popup scores are not related to task similarity. They simply indicate what parameters should be included in the sub-network for the current task. As a result, the popup scores are not necessarily closer for similar tasks because there could be many possible sub-networks even for the same task. Our method only finds one. The initialization and soft masking are mainly used for dealing with forgetting and knowledge transfer. They only indirectly affect the current popup scores as those parameters/knowledge that can be shared across tasks should have higher popup scores. They will not affect those already learned popup scores for previous tasks because we save the sub-networks identified by popup scores for previous tasks (L308).
>
> > Q8: why was there no specific explanation for the chosen value of the threshold parameter (ε)?  Is this parameter set uniformly for all tasks, or do different tasks have different settings?
>
> Since we proposed a novel method to enable the supermask method [9] to transfer knowledge across tasks, which originally learns each task completely independently, we follow [9] to set the value of the threshold. The same threshold is used across all tasks
>
> [9]: Supermasks in Superposition, Wortsman et al., NeurIPS 2020
>
> > Q9: what is the calculation method for normalizing and rounding the gradient-based importance using Tanh()? Why do the importance scores after Tanh() activation fall within the [0, 1] range, despite the original range being [-1, 1]? What impact does normalizing the importance values to -μ for parameters that initially had an importance of 0 have on the subsequent importance calculations?
>
> We apply normalization to the importance of each layer, i.,e., $\text{Norm}(I_l^{(k)})= \frac{I_l^{(k)} - I_l^{(k)}.mean()}{I_l^{(k)}.std()}$. We noticed that we had a typo in Eq. 6. The absolute operation should not be taken in Eq. 6 but after the Tanh so that the importance scores fall within [0,1]. Let the “initial importance” be the importance score computed after (the corrected) Eq. 6. The normalized importance for the 0 initial importance is around 0 (may not be exactly 0) because the mean is always close to 0. As a result, $Norm(I_l^{(k)})\approx 0$ and its corresponding parameter can be updated with almost no masking (Eq.5) by the subsequent task.
>
> > Q10: since different tasks have different evaluation metrics, why did they choose to average F1, BLEU, and ROUGE-1 scores? Why not normalize task performances before averaging?
>
> We reported the average score mainly for illustration purposes as the range of all metrics fall in [0, 1]. We agree that the readers should compare the individual results. We see the TSS works the best in terms of individual results as well.
>
> > Q11:  why did the authors use ONE as a control for analyzing Knowledge Transfer (KT) when ONE and TSS have different model architectures? Wouldn't it be more reasonable to compare each task's performance independently with its performance in the task sequence?
>
> All systems (including ONE) use the same backbone BART-large. We use ONE as control because ONE trains model independently, with no transfer or forgetting.
>
> We believe your second question is about comparing TSS with TSS training each task independently. TSS training each task independently is the same as removing soft-masking (which is TSS (w/o SM) in Table 2). Table 2 shows that TSS outperforms “TSS (w/o SM)”, which also indicates forward knowledge transfer.
>
> > Q12: Missing Reference: Konishi, T., Kurokawa, M., Ono, C., Ke, Z., Kim, G., & Liu, B. (2023). Parameter-Level Soft-Masking for Continual Learning. arXiv preprint arXiv:2306.14775.
>
> This paper was published in ICML-2023, which took place (July 23-29, 2023) after the EMNLP submission deadline. Our method is also different. The main difference is that their method cannot guarantee no forgetting as their soft-masking allows changes to parameters of previous tasks. On the contrary, TSS guarantees no forgetting with the help of sub-network discovery. What’s more, their soft-masking is designed only for classification tasks, which is not applicable to generation tasks like SUM and DRG. Not to mention it is also designed for image datasets. We will include a discussion about the paper in our revision.
>
> > Q13: In Equation (3), the subscripts on the left and right sides are inconsistent, and the variable definition for w_{i,j} has not been declared.
>
> We didn’t explicitly declare $w_{i,j}$ because we thought it can be inferred from $s_{i,j}$. $s_{i,j}$ means a popup score in layer $l$ and similarly, $w_{i,j}$ means a weight (in our case, it is the adapter weight, which is always fixed) in layer $l$.

---

### Official Review · Reviewer_dnuC · 2023-08-05

**Soundness:** 4

**Excitement:**

4: Strong: This paper deepens the understanding of some phenomenon or lowers the barriers to an existing research direction.

**Paper Topic And Main Contributions:**

The paper's area is Continual Learning (CL), where a model is supposed to solve a sequence of similar or not-quite-similar tasks. The authors propose a new way to train such a model. The approach is designed in order to overcome two common Continual Learning problems: catastrophic forgetting (when the model "forgets" how to solve the previously seen tasks) and limited knowledge transfer (when the model's "experience" of solving tasks does not help it to solve a new task).

Contributions:
* New continual learning algorithm which try to overcome the problems faced by all other algorithms in the field.
* The core idea of the method (as far as I understood) is based on "attention", which is quite a popular and wide-used concept.
* Detailed description of the area and approaches previously used (not a survey study but sometimes feels like it).

**Reasons To Accept:**

Reasons To Accept:
* New continual learning algorithm which try to overcome the problems faced by all other algorithms in the field.
* Nice algorithm scheme illustrations.
* Extensive experiments with many models and datasets.
* Ablation study.

**Reasons To Reject:**

Except for minor linguistic/presentation flaws, I see no reason to reject the paper.

**Reproducibility:**

4: Could mostly reproduce the results, but there may be some variation because of sample variance or minor variations in their interpretation of the protocol or method.

**Reviewer Confidence:**

2: Willing to defend my evaluation, but it is fairly likely that I missed some details, didn't understand some central points, or can't be sure about the novelty of the work.

**Typos Grammar Style And Presentation Improvements:**

* Line 22: "AI". Probably also needs full naming when abbreviation is first used.
* Line 34 (also line 178): "should (1) overcome". It took some time before I realized the meaning of number "(1)". If this is arranged as a paragraph (not a numbered list), then maybe better use words like "first", "second".
* Line 123: Better say "real number", not just "number".
* Figure 1: Right column's captions not centered (forward propagation, backward propagation).
* Line 333: "Importance values" (plural).
* Line 336: "as follows." (point not colon).
* Footnote 7: duplicate "the"
* Line 367: "Kaiming Initialization". Provide a reference?
* Line 378: "pupup scores"
* Line 380 (eq. 6): Small vertical braces. Also, maybe something is wrong with the denominator.
* Lines 446 - 474 (and many others). I highly recommend reconsidering the use of bold fony. In my opinion, currently it is used too much.
* Line 460: excess "dot" symbol.
* Table 2: not all best values in "FR" column are typed on bold.
* Line 527: "Knowledge transfer (KT)" In my opinion, better to omit "(KT)" here. (Or, on the contrary, omit "Knowledge transfer")
* Line 993: "Let us use some examples to illustrate:" (colon, not comma). Also, probably better use a numbered list below, not bulleted.
* Lines 1003, 1004: Wrong formatting of negative numbers (this is not a minus sign).
* Table 12: Training time 48000 min... Maybe I missed something but such an ominous number just stunned me for a while :)

---

> ### Author Rebuttal · Authors · 2023-08-29
>
> Thank you very much for appreciating our work and for your careful reading. We will fix the typos, grammar style and presentation issues in the revised version.

---

### Meta-Review · Area_Chair_v3F3 · 2023-09-19

**Recommendation:** 4

**Metareview:**

The authors proposed a new methodology for continual learning, which introduces an importance-based gradient soft-masking on top of SupSup to enhance transfer capacity. The paper itself in general is well-structured and easy to understand. In terms of weaknesses, there was disagreement among the reviewers ranging from the presentation only having minor presentation flaws, to criticism about the applicability of the proposed method, concerns about the impact of reduced data on the baselines results that needs to be further discussed, and one reviewer arguing lack of novelty, that the methodology was hard to follow, and raising that results concerning improved transfer ability and replay-based methods are not shown in the experiments. Moreover, one reviewer suggested the authors may consider including results for settings with resource disparities. In response, the authors clarify their setting and its applicability, addressed the criticism about the impact of the size of the dataset on the baselines and also that about resource disparities, provided additional experiments including a replay-based method, and addressed concerns related to novelty, clarity and provided additional results to highlight the improved transfer capabilities of the proposed methodology. It is worth noting that the authors presented a comprehensive rebuttal including detailed responses to the reviewers concerns and questions and included substantial new experimental results. Moreover, the reviewers who raised concerns engaged in discussion with the authors. However, one of the reviewers still insisted that the proposed approach is incremental and of limited value to the NLP community.

---

### Decision · Program_Chairs · 2023-10-07

**Decision:**

Accept-Findings

**Comment:**

The authors proposed a new methodology for continual learning, which introduces an importance-based gradient soft-masking on top of SupSup to enhance transfer capacity. The paper itself in general is well-structured and easy to understand. In terms of weaknesses, there was disagreement among the reviewers ranging from the presentation only having minor presentation flaws, to criticism about the applicability of the proposed method, concerns about the impact of reduced data on the baselines results that needs to be further discussed, and one reviewer arguing lack of novelty, that the methodology was hard to follow, and raising that results concerning improved transfer ability and replay-based methods are not shown in the experiments. Moreover, one reviewer suggested the authors may consider including results for settings with resource disparities. In response, the authors clarify their setting and its applicability, addressed the criticism about the impact of the size of the dataset on the baselines and also that about resource disparities, provided additional experiments including a replay-based method, and addressed concerns related to novelty, clarity and provided additional results to highlight the improved transfer capabilities of the proposed methodology. It is worth noting that the authors presented a comprehensive rebuttal including detailed responses to the reviewers concerns and questions and included substantial new experimental results. Moreover, the reviewers who raised concerns engaged in discussion with the authors. However, one of the reviewers still insisted that the proposed approach is incremental and of limited value to the NLP community.